# Structure and function of a family of tick-derived complement inhibitors targeting properdin

Katharina Braunger [1,7], Jiyoon Ahn[1,7], Matthijs M. Jore [1,5,7], Steven Johnson[1,2✉], Terence T. L. Tang[1,6], Dennis V. Pedersen[3], Gregers R. Andersen [3] & Susan M. Lea [1,2,4✉]

Activation of the serum-resident complement system begins a cascade that leads to activation of membrane-resident complement receptors on immune cells, thus coordinating serum and cellular immune responses. Whilst many molecules act to control inappropriate activation, Properdin is the only known positive regulator of the human complement system. By stabilising the alternative pathway C3 convertase it promotes complement self-amplification and persistent activation boosting the magnitude of the serum complement response by all triggers. In this work, we identify a family of tick-derived alternative pathway complement inhibitors, hereafter termed CirpA. Functional and structural characterisation reveals that members of the CirpA family directly bind to properdin, inhibiting its ability to promote complement activation, and leading to potent inhibition of the complement response in a species specific manner. We provide a full functional and structural characterisation of a properdin inhibitor, opening avenues for future therapeutic approaches.

[1] Sir William Dunn School of Pathology, University of Oxford, OX1 3RE Oxford, UK. [2] Center for Structural Biology, Center for Cancer Research, National Cancer Institute, 21702 Frederick, MD, USA. [3] Department of Molecular Biology and Genetics, Aarhus University, DK-8000 Aarhus, Denmark. [4] Central Oxford Structural Molecular Imaging Centre, University of Oxford, OX1 3RE Oxford, UK. [5] Present address: Department of Medical Microbiology, Radboud University Medical Centre, Nijmegen, Netherlands. [6] Present address: MRC Laboratory of Molecular Biology, Cambridge, UK. [7] These authors contributed equally: Katharina Braunger, Jiyoon Ahn, Matthijs M. Jore. ✉email: steven.johnson2@nih.gov; susan.lea@nih.gov

The complement system plays pivotal roles in immunity and cellular homeostasis[1–3]. The tightly regulated proteolytic cascade can be activated through the Classical pathway (CP), Lectin pathway (LP), and Alternative pathway (AP), all of which ultimately lead to the formation of surface-bound C3 convertases that cleave C3 into C3a and C3b. This induces a self-amplification loop involving the AP convertase C3bBb, leading to opsonisation with C3b and the release of the anaphylatoxin C3a. Exceeding a certain density of surface bound C3b triggers a shift in substrate preference towards C5[4]. The resulting production of the anaphylatoxin C5a, and the first component of the membrane attack complex (MAC) C5b, initiate the terminal steps of the cascade.

A plethora of host proteins are known to attenuate the response at all stages of the pathway, including inhibition of the C1 protease by C1-inhibitor, dissociation/degradation of convertases by regulators such as factor H and factor I, and prevention of MAC insertion by CD59. Such regulation is important as excessive, inappropriate, or prolonged complement activation can be highly detrimental to host tissue[5,6]. A striking example is constituted by the disproportionate immune response seen in many severe COVID-19 cases of the ongoing pandemic, with raised complement markers including C5a and soluble MAC observed in patients[7–10]. In contrast, the only identified positive regulator in the complement system is properdin[11–13].

Properdin natively forms oligomers (Supplementary Fig. 1) which promote assembly of the C3bB proconvertase and stablise the AP C3 convertase, thereby extending its half-life 5- to 10-fold and inhibiting its degradation by factor I[12,14,15]. Besides this, there is an ongoing controversy about properdin's potential role as a pattern recognition molecule, in which it is suggested to bind directly to activating surfaces, thereby recruiting C3 and inducing complement activation[16,17].

Properdin is a highly flexible protein consisting of an N-terminal TGFβ binding (TB)-domain followed by six thrombospondin type I repeats (TSR)-domains (Supplementary Fig. 1). The convertase binding region lies in a vertex formed between two properdin molecules, interacting head-to-tail. High resolution structures of pseudo-monomeric properdin in isolation and bound to the C-terminal domain of C3b, as well as a moderate resolution structure of a properdin-bound AP C3 convertase, have greatly improved our understanding of properdin multimerization and convertase stabilisation[18–20].

Properdin inhibition leading to complement downregulation has the potential to provide an alternative therapeutic strategy with different downstream effects to targeting other points of the pathway. By acting specifically at the level of the AP C3 convertases, properdin inhibition leads to a reduction in the production of both C3a and C5a, reduction in C3b opsonisation, and reduced MAC formation, but has the potential to preserve some non-amplified complement activation via CP and LP. Additionally, properdin concentration in serum (5–25 μg/mL) is much lower than for example C3 (1.2 mg/mL) or C5 (75 μg/mL), such that a lower dose of an inhibitory agent might be sufficient to demonstrate therapeutic effects[21].

Tick-saliva presents a vast source for pharmaco-active proteins, targeting a wide array of immune mechanisms[22]. Their therapeutic potential can be illustrated with the example of Nomacopan (Coversin, Akari therapeutics), a recombinant variant of the lipocalin family protein OmCI (Ornithodoros moubata Complement Inhibitor), which is currently under investigation in several clinical trials at various stages. Its mechanism of action is binary, with tight binding of C5 inhibiting the terminal pathway of complement[23,24], and sequestration of the proinflammatory eicosanoid leukotriene B4 (LTB4) within an internal binding cavity[25] providing additional anti-inflammatory function.

In this study, we identify a family of complement inhibitors from the hard tick Rhipicephalus pulchellus, hereafter termed CirpA (Complement inhibitor from R. pulchellus of the alternative pathway). CirpA1 targets human properdin, leading to potent AP inhibition. Functional analysis of six CirpA homologs demonstrates a highly species dependent activity profile. In addition, we present crystal structures of four CirpA homologs, revealing a conserved lipocalin fold, and the crystal structure of the properdin-CirpA1 complex. Our work represents a comprehensive functional and structural characterisation of a properdin inhibitor. The structural and functional analysis gives insights into the mechanism of properdin binding and reveals a variety in lipocalin inhibition mechanisms.

## Results

**Tick saliva protein CirpA1 targets human Properdin.** Fractionation of R. pulchellus salivary gland extract (SGE) had previously led to identification of the CirpT family of complement inhibitors[26]. Strikingly, the first chromatographic step indicated a distinct second population with inhibitory activity specific for the alternative pathway of complement activation (Supplementary Fig. 2a). Using an analogous procedure, we further purified and enriched fractions with inhibitory activity, demonstrated by reduced MAC deposition in a standard complement alternative pathway activation assay using human serum (Fig. 1a, b; Supplementary Fig. 2a, b).

The proteins in the active fractions were analysed by electrospray ionisation-MS/MS following trypsin digest. Analysis against two published cDNA databases[26,27] resulted in a list of nine proteins containing a predicted signal peptide (Supplementary Fig. 2c). The candidates were expressed recombinantly in Drosophila melanogaster S2 cells and the culture supernatants tested for complement inhibitory activity. One protein, subsequently termed CirpA1, was shown to inhibit AP activation of the human complement system, while having no significant impact on the CP or LP at the concentrations tested (Fig. 1c and Supplementary Fig. 2d). This result is especially interesting given the role of the AP in the amplification loop that boosts both the CP and LP. CirpA1 significantly inhibited the formation of both C3a and C5a via the alternative pathway (Fig. 1d, e) suggesting it acts at the point of C3 cleavage.

To pinpoint the target of CirpA1, we performed a pull-down assay from human serum utilising immobilised CirpA1. Western Blot analysis using antibodies against alternative pathway-specific proteins resulted in properdin being identified as a CIRpA1 binding protein (Fig. 1f and Supplementary Fig. 3). Properdin is the only known positive regulator of the human complement system. In the alternative pathway, properdin binds to the C-terminal domain of C3b in the C3 convertase (C3 C345c), thereby increasing convertase half-life approximately ten-fold. To gain a better understanding how CirpA1 binding to properdin leads to inhibition of the alternative pathway of complement activation we investigated how it interferes with properdin function. We generated biotinylated C3b from purified C3 to couple it to streptavidin magnetic beads in a physiological orientation[4]. We incubated the C3b coupled beads in human serum, supplemented with EDTA and EGTA to block complement activity, in presence or absence of CirpA1. Subsequent western blot analysis clearly shows that CirpA1 abolishes properdin-C3b binding. Furthermore, CirpA1 is able to act in a competitive manner, releasing pre-bound properdin (Fig. 1g).

**The CirpA family of inhibitors show species specificity.** To identify potential biologically relevant homologs, the CirpA1 sequence was used to query the expressed sequence tag database (NCBI), as

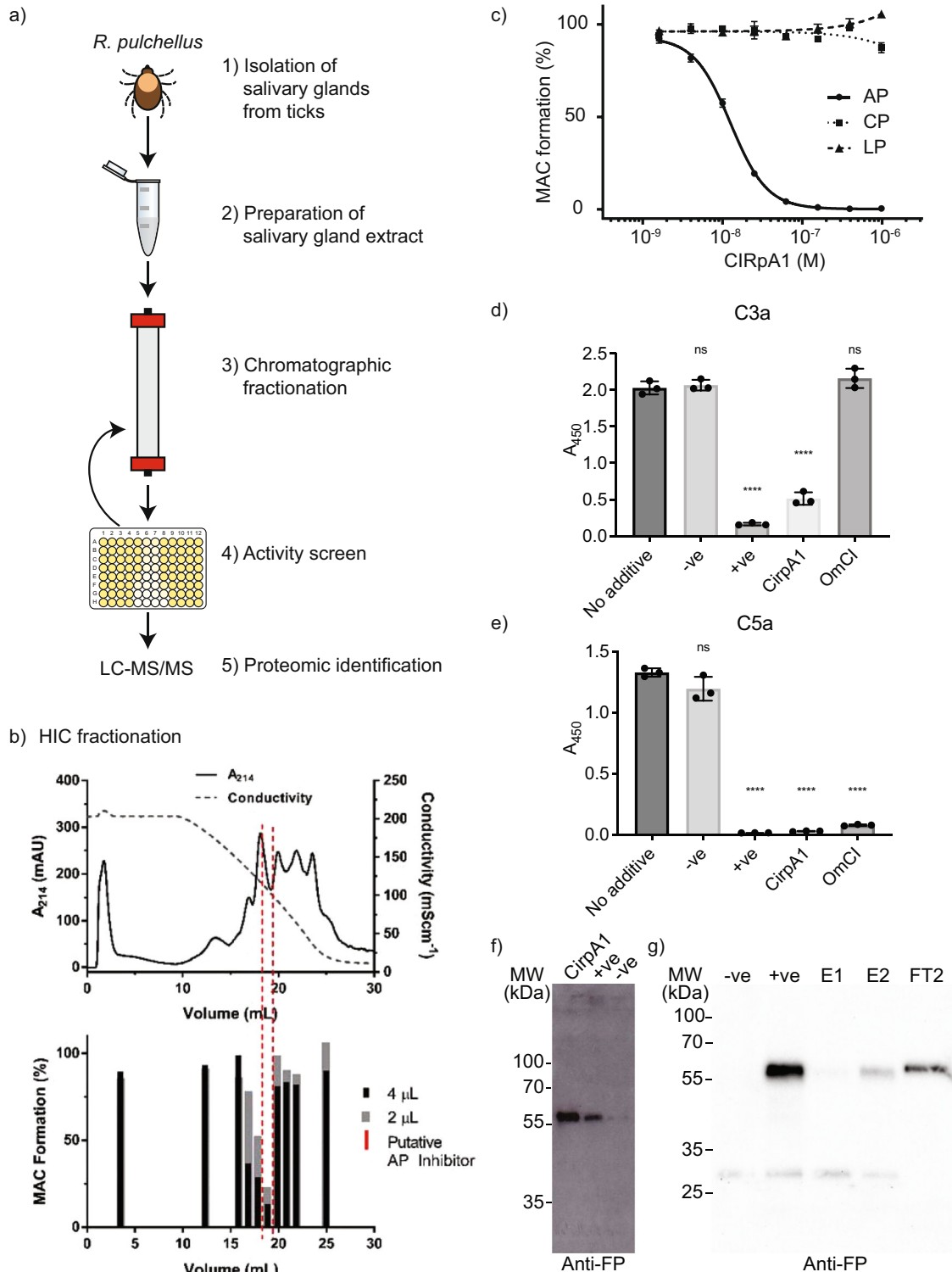

well as in-house *R. appendiculatus*[28] and *R. pulchellus* sialomes. This search revealed five homologs across different tick species (Fig. 2a and Supplementary Fig. 4a) with varying sequence identity (hereafter named CirpA2-A6, pairwise sequence identity to CirpA1 43–82% over ~200 residues). To compare their activity all CirpA homologs were expressed in *D. melanogaster* S2 cells and assessed regarding their ability to inhibit alternative pathway complement mediated haemolysis via sera from multiple mammalian species.

We tested inhibitory activity against human, monkey, rat, and guinea pig serum (Fig. 2b). CirpA1 showed comparable activity in

human and monkey sera ($IC_{50} = 1.41 \times 10^{-8}$ and $1.38 \times 10^{-8}$ respectively) but only weak effects in rat serum and was inactive in guinea pig. Interestingly, for CirpA2, which shares 82% of amino acids with CirpA1 and only differs in the C-terminal 57 residues, the inhibitory potential is lost in human and strongly reduced in monkey. In contrast, CirpA6, the homolog with the lowest sequence identity to CirpA1 (43%), shows almost identical inhibition behaviour in human ($IC_{50} = 1.40 \times 10^{-8}$), considerable activity in monkey and weak inhibition in rat. The homologs CirpA3-5 were inactive in the species tested in this study (Fig. 2b).

**Fig. 1 Identification and functional characterisation of CirpA1 from *Rhipicephalus pulchellus* salivary glands. a** Experimental procedure leading to the identification of CirpA1. **b** Fractionation of salivary gland extract hydrophobic interaction chromatography (top) used to identify fractions with inhibitory activity against the alternative pathway in a AP haemolysis assay (bottom). **c** Inhibition of complement pathways by CirpA1 measured in WIESLAB assays using human serum. CirpA1 specifically inhibited the alternative pathway. Values were normalised for no-serum samples (0% MAC formation) and no-additive samples (100% MAC formation). Error bars, s.e.m. from three independent experiments ($n = 3$). Curve fitting was carried out in GraphPad Prism using a dose response inhibition (variable slope) model. AP $IC_{50} = 13$ nM. C3a (**d**) and C5a (**e**) levels in supernatants of the alternative pathway Wieslab assay. CirpA1 inhibited C3a and C5a formation through the alternative pathway; −ve = Ra-HBP2 (non-complement inhibitor); +ve = EDTA (general complement inhibitor). The classical pathway inhibitor OmCI was added for comparison. All proteins were added at a final concentration of 1 µM. Values were baseline-corrected for buffer-only samples. Error bars, s.d. of the mean ($n = 3$ Wieslab samples). ns (not significant); ****$P < 0.001$ by unpaired two-tailed t-test, with no-additive sample as the reference. **f** CirpA1 binds properdin, anti-properdin blot using CirpA1 coated beads from serum compared to +ve (known properdin binder, *Ixodes scapularis* Salp20) and −ve controls (empty beads). Representative result of $n = 5$. **g** Anti-properdin Western Blot after serum incubation of C3b-coupled beads in presence or absence of CirpA1 shows CirpA1 is able to prevent binding to C3b (E1) and displace the majority of pre-bound properdin (E2, FT2); −ve = no added serum; +ve = no inhibitor. Representative result of $n = 3$. Source Data are provided as a Source Data file.

**CirpA proteins display a conserved lipocalin fold**. To understand the distinct inhibition profiles, we set out to determine the structure of the different CirpA homologs. To that end we over-expressed the proteins in *Escherichia coli*, purified them via refolding from inclusion bodies, and confirmed that the material retained equivalent secondary structure and biological activity to the insect cell produced material (Supplementary Fig. 5). Using this strategy, we were able to purify CirpA1, A3, A4, and A5 and determine their structure using X-ray crystallography to a resolution of 2.0, 2.1, 1.8, and 1.9 Å respectively. (Fig. 2c, d and Table 1).

Sequence analysis of CirpA1, using the FFAS server (https://ffas.godziklab.org/ffas-cgi/cgi/ffas.pl), revealed very weak homology to the C5-binding complement inhibitor OmCI (sequence identity of 14%). The structure of CirpA1 was solved by molecular replacement using the tick-derived complement inhibitor OmCI (pdb-id: 3ZUI) as an initial search model. The refined model of CirpA1 was used to determine the structures of CirpA3-5. All CirpA structures share a lipocalin fold with a short N-terminal α-helix followed by an eight-stranded beta-barrel and a longer C-terminal α-helix (Supplementary Fig. 4b–f). Overall, the structures appear very similar, with the biggest conformational differences in the tilt angle of the C-terminal helix H2 as well as variations in loops between beta strands 4 and 5 (L4-5) and between strands 7 and 8 (L7-8).

**CirpA binds properdin at the TSR5-6 junction**. We next aimed to get more detailed insights into the mechanism of function of CirpA1 to potentially understand differences in inhibitory potential between the homologs. We first investigated whether properdin was still able to form multimers upon binding of CirpA1. Size exclusion analysis of purified properdin in presence of CirpA1 resulted in clear shifts towards higher molecular weight for all properdin populations with no noticeable changes in population ratio (Fig. 3a). Hence, we concluded that CirpA1 binds all multimeric forms of properdin without affecting multimer distribution.

Next, we set out to determine the structure of CirpA1-properdin. We chose to work with a minimal properdin construct that had previously been used to determine the crystal structure of pseudo-monomeric, minimal functional unit of properdin (hereafter referred to as FPΔ2,3)[18,19]. It consists of a head-fragment with the TB domain, as well as TSR1 and a tail fragment of TSR 4-6. Co-expression of both fragments in HEK293F cells results in the formation of a vertex mimicking properdins physiological multimerization behaviour and is able to bind the C3 convertase[18]. The addition of CirpA1 to purified FPΔ2,3 results in a distinct shift of molecular weight (Fig. 3b), as measured by SEC-MALS, indicating the formation of a complex

at 1:1 stoichiometry. In addition, we carried out micro-scale thermophoresis to assess the binding affinity of the interaction, demonstrating a $K_d$ of 97 nM (Supplementary Fig. 6).

We were able to obtain crystals from the FPΔ2,3-CirpA1 complex and determine its structure to 3.4 Å by X-ray crystallography. The structure was solved by molecular replacement, using models of CirpA1 (this study) and FPΔ2,3 (pdb-id: 6S08) as search models. In the crystal lattice each FPΔ2,3 molecule interacts with two inhibitor molecules (Fig. 3c). Knowing from our SEC-MALS analysis that each FPΔ2,3 binds only one CirpA1 in solution, we designed two CirpA1 mutants, introducing arginine residues at distinct positions in each interface (E170R/V173R and Q148R) which based on our structure would likely disrupt the corresponding interaction with properdin. Using an alternative pathway haemolysis assay, we were able to show that CirpA1 E170RV173R displays similar properties as the wild-type (Fig. 3d). In contrast, the CirpA1 inhibitory effect is reduced in the Q148R mutant indicating that interface 2 is the functionally relevant one. CirpA1 binds properdin at TSR5 and 6, in immediate proximity to the convertase binding region (Fig. 3e). We observe clear density for the majority of large side chains resulting in high confidence in our fitted model (Fig. 3f).

Analysis of the interaction surface, which is burying 864 Å², reveals complementary charged patches, suggesting that binding is mediated by electrostatic interactions (Fig. 4a). Superposition with structures of isolated FPΔ2,3 as well as FPΔ2,3 in complex with the C-terminal domain of C3b shows subtle conformational changes in the binding region (Fig. 4b). In CirpA1, properdin binding leads to structural changes in the three C-terminal beta strands, as well as specific rearrangements of individual amino acids contributing to complex stability (Fig. 4c). One example is Q148 which forms a hydrogen bond to properdin Q338 (Figs. 3f and 4c). Superposition with structures of the CirpA3-5 shows that CirpA3 and CirpA4 could rearrange analogously to form this interaction, while in CirpA5 Q148 is replaced with a glycine residue (Fig. 4d). Interestingly, sequence comparison of properdin between the species tested for CirpA activity shows that Q/E at position 338 is unique to primates. Our structure suggests that substitution with aspartate (rat) or lysine (guinea pig) would likely weaken the interaction, contributing to CirpA1 activity loss in these species (Fig. 4e).

Another CirpA1 residue undergoing a conformational rearrangement upon properdin binding is Y122 which forms a hydrophobic pocket around properdin P377 together with CirpA1 residues F122, Y131, and F150. This pocket is incomplete in all three CirpA1 homolog structures.

Together with the fact that the CirpA1 regions forming the properdin binding interface overlap with the regions of highest conformational variability between the CirpA homolog structures

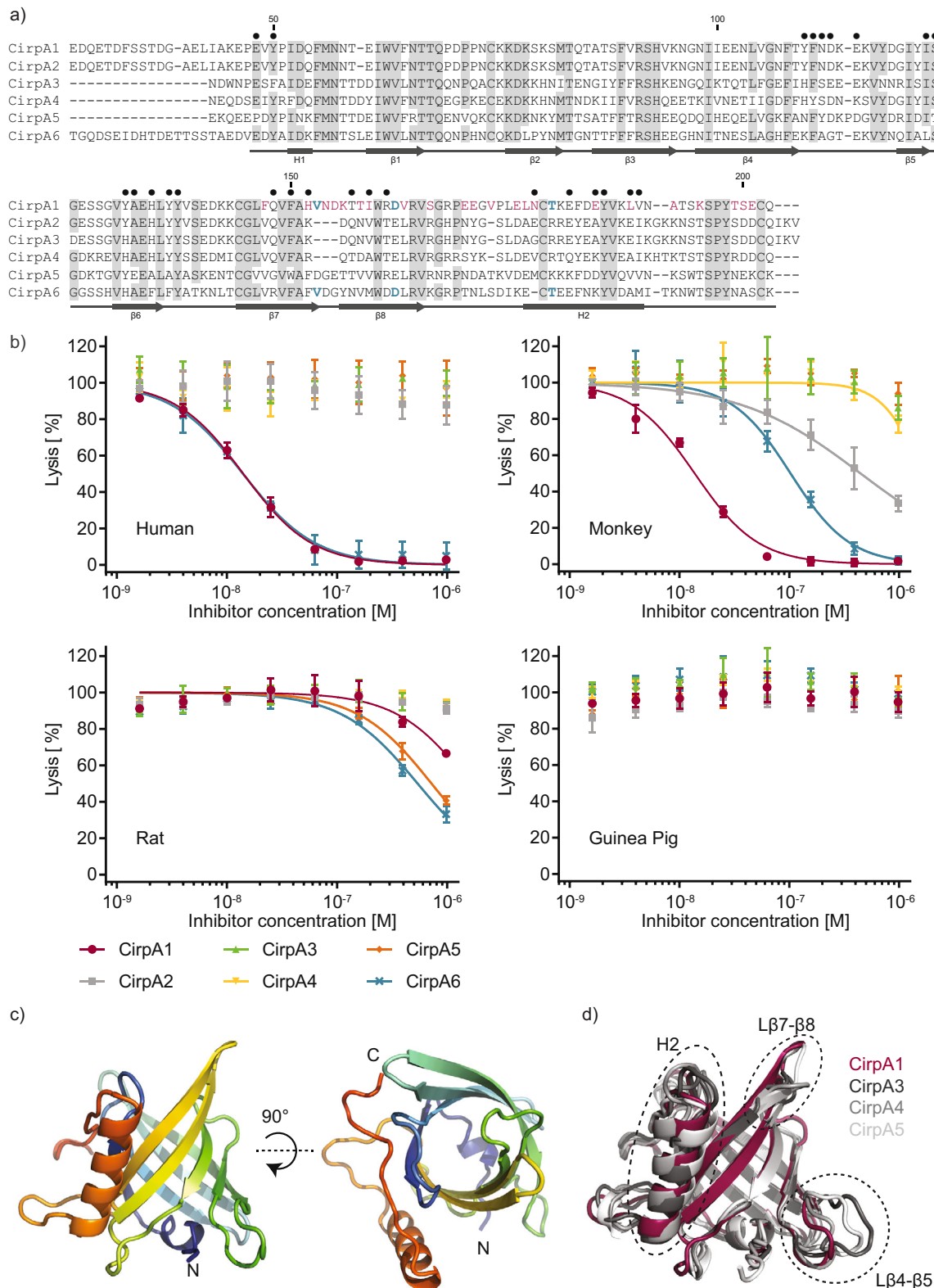

(Fig. 4f), these changes might provide an explanation for the lack of human properdin inhibition activity of CirpA3-A5 in our analysis.

In a broader context of complement activation, CirpA1 is not the first known tick lipocalin mediating potent inhibition. The complement inhibitor OmCI binds the C5 in close vicinity to its C345C domain and prevents cleavage of C5 by the C5 complement convertases, thereby preventing the release of anaphylatoxin C5a and formation of the terminal MAC (Fig. 5a). In contrast, CirpA1 acts at an upstream step, by binding to properdin and interfering with assembly and stabilisation of the C3 convertase. Despite their high structural similarity (r.m.s.d. of 1.4 Å over 105 Cαs), OmCI and CirpA1 bind their targets in very different regions of the lipocalin surface (Fig. 5b–d). In addition,

**Fig. 2 Characterisation of the CirpA protein family. a** Sequence alignment of CirpA family members. CirpA1 was purified from R. pulchellus, CirpA2 and CirpA3 are homologs from the *R. pulchellus* transcriptome (Reichhardt 2020), CirpA4: GenBank CD794868.1, CirpA5 was identified from the *R. appendiculatus* transcriptome (Jore 2016), CirpA6: GenBank CK182034.1; Grey: residues conserved in five or more, blue: residues unique to homologs with anti-complement AP activity in Fig. 2b, red: residues unique to CirpA1, black dots: residues involved in the CirpA1- FPΔ2,3 interface in the complex structure in Fig. 3e according to PISA analysis. **b** AP haemolysis assay with serum from *Homo sapiens* (human), *Macaca fascicularis* (monkey), *Rattus norvegicus* (rat), and *Cavia porcellus* (guinea pig). Values were normalised for no-serum samples (0% lysis) and no-additive samples (100% lysis). Error bars, s.e.m. from three independent experiments ($n = 3$). Curve fitting was carried out in GraphPad Prism using a dose-response inhibition (variable slope) model. **c** Crystal structure of CirpA1. **d** Overlay of crystal structures of CirpA1, CirpA3, CirpA4, and CirpA5. Regions of high conformational variability between homologs indicated by dashed ellipsoids. Source Data are provided as a Source Data file.

**Table 1 X-ray crystallographic data and model quality.**

|  | CirpA1 | CirpA3 | CirpA4 | CirpA5 | CirpA1-FPΔ2,3 |
|---|---|---|---|---|---|
| Data collection |  |  |  |  |  |
| Space group | $P222_1$ | $P2_1$ | $P2_12_12_1$ | $I222$ | $P2_12_12$ |
| $a, b, c$ (Å) | 72.6, 45.8, 57.5 | 68.4, 150.3, 68.4 | 40.1, 54.3, 79.7 | 52.0, 71.3, 110.2 | 166.3, 54.5, 70.6 |
| $\alpha, \beta, \gamma$ (°) | 90.0, 90.0, 90.0 | 90.0, 91.83, 90.0 | 90.0, 90.0, 90.0 | 90.0, 90.0, 90.0 | 90.0, 90.0, 90.0 |
| Resolution (Å) | 45.06-1.96 (2.01-1.96) | 62.25-1.92 (2.11-1.92) | 44.90-1.83 (1.86-1.83) | 55.08-1.91 (1.96-1.91) | 83.16-2.84 (3.28-2.84) |
| $R_{sym}$ or $R_{merge}$ | 12.2 (131.7) | 12.0 (67.4) | 7.0 (47.9) | 24.7 (180.0) | 35.7 (182.0) |
| $I/\sigma I$ | 11.5 (1.4) | 6.3 (1.5) | 11.0 (2.1) | 5.3 (1.0) | 6.1 (1.7) |
| Completeness (%) | 100.0 (100) | 92.2 (61.4) | 93.2 (100.0) | 99.9 (99.9) | 89.4 (62.5) |
| Redundancy | 11.4 (7.1) | 3.3 (3.2) | 4.0 (3.7) | 6.4 (6.9) | 10.7 (9.7) |
| $CC_{1/2}$ (%) | 99.8 (57.1) | 99.3 (51.5) | 99.7 (77.7) | 98.8 (58.0) | 99.4 (59.9) |
| Refinement |  |  |  |  |  |
| Resolution (Å) | 45.07-1.96 (2.11-1.96) | 62.25-2.10 (2.14-2.10) | 44.90-1.83 (1.97-1.83) | 55.08-1.91 (2.03-1.91) | 24.71-3.40 (4.28-3.40) |
| No. reflections | 14297 | 60529 | 14798 | 16258 | 6822 |
| $R_{work}/R_{free}$ | 19.3/21.9 | 21.7/25.0 | 19.2/23.6 | 22.0/24.4 | 22.9/27.4 |
| *No. atoms* |  |  |  |  |  |
| Protein | 1273 | 9935 | 1305 | 1340 | 3684 |
| Ligand/Ion | 18 | 5 | 4 | 15 | 120 |
| Water | 116 | 711 | 161 | 167 | - |
| *B-factors* |  |  |  |  |  |
| Protein | 39.0 | 34.2 | 28.8 | 31.1 | 65.2 |
| Ligand/Ion | 70.4 | 37.8 | 63.0 | 78.7 | 62.7 |
| Water | 40.5 | 28.7 | 36.4 | 7.4 | - |
| *R.m.s. deviations* |  |  |  |  |  |
| Bond length (Å) | 0.008 | 0.003 | 0.004 | 0.002 | 0.004 |
| Bond angle (°) | 0.887 | 0.531 | 0.706 | 0.534 | 0.874 |

*Values in parentheses are for highest-resolution shell.

OmCI displays a second mechanism of function by binding the proinflammatory eicosanoid leukotriene B4 (LTB4) within an internal binding cavity. The corresponding pocket in CirpA1 is blocked by a cluster of charged residues, precluding binding of a hydrophobic ligand in this position (Fig. 5e, f).

## Discussion

We have identified a complement inhibitor from the hard tick *R. pulchellus*. Our functional and structural analyses show that it targets properdin, the only known positive regulator of the human complement system. Sequence analyses identified five additional members of the CirpA family, with pairwise sequence identities to CirpA1 between 43–82%. Their anti-complement activity does not seem to correlate with the degree of sequence conservation and is specific to serum of certain species. To understand the differences in inhibition profile, we determined the structures of CirpA1, CirpA3, CirpA4, CirpA5, and CirpA1-FPΔ2,3, which confirmed a lipocalin-like fold for all CirpA homologs. We identified three regions with high conformational variability between homologs. All these regions are part of the observed binding interface between CirpA1 and properdin, and hence the conformational differences might render other homologs incompatible with binding human properdin. However, it is

likely that the homologues have evolved to target properdin from other species, especially given the identification of multiple CirpA homologues in a single tick species. *R. pulchellus* and *R. apendiculatus* are both "three host" ticks that feed on multiple hosts during a lifecycle, and therefore need the capacity to inhibit multiple host immune systems. We would predict that some of the homologues would target ungulates, as these are the major natural hosts of these ticks.

We also cannot rule a role for CirpA homologues in regulating other pathways. Lipocalins are highly versatile and functionally diverse[29,30], as illustrated by our comparison of CirpA1 with another tick based complement inhibitor OmCI. OmCI has a dual role in modulating the host immune response, being able to inhibit C5 cleavage via direct interaction with C5 while simultaneously sequestering leukotriene B4 in the lipocalin hydropobic cavity. Our CirpA1- FPΔ2,3 structure demonstrates that complement inhibition is achieved not only by binding a different family of complement protein, but also using a completely different surface of the lipocalin fold. In addition, the CirpA family lacks the hydrophobic pocket seen in many lipocalins, suggesting that their mechanism of action is likely to be exclusive to their capacity to form protein–protein interactions.

The CirpA1- FPΔ2,3 structure also sheds light on how CirpA1 interferes with properdin function. High resolution structures of

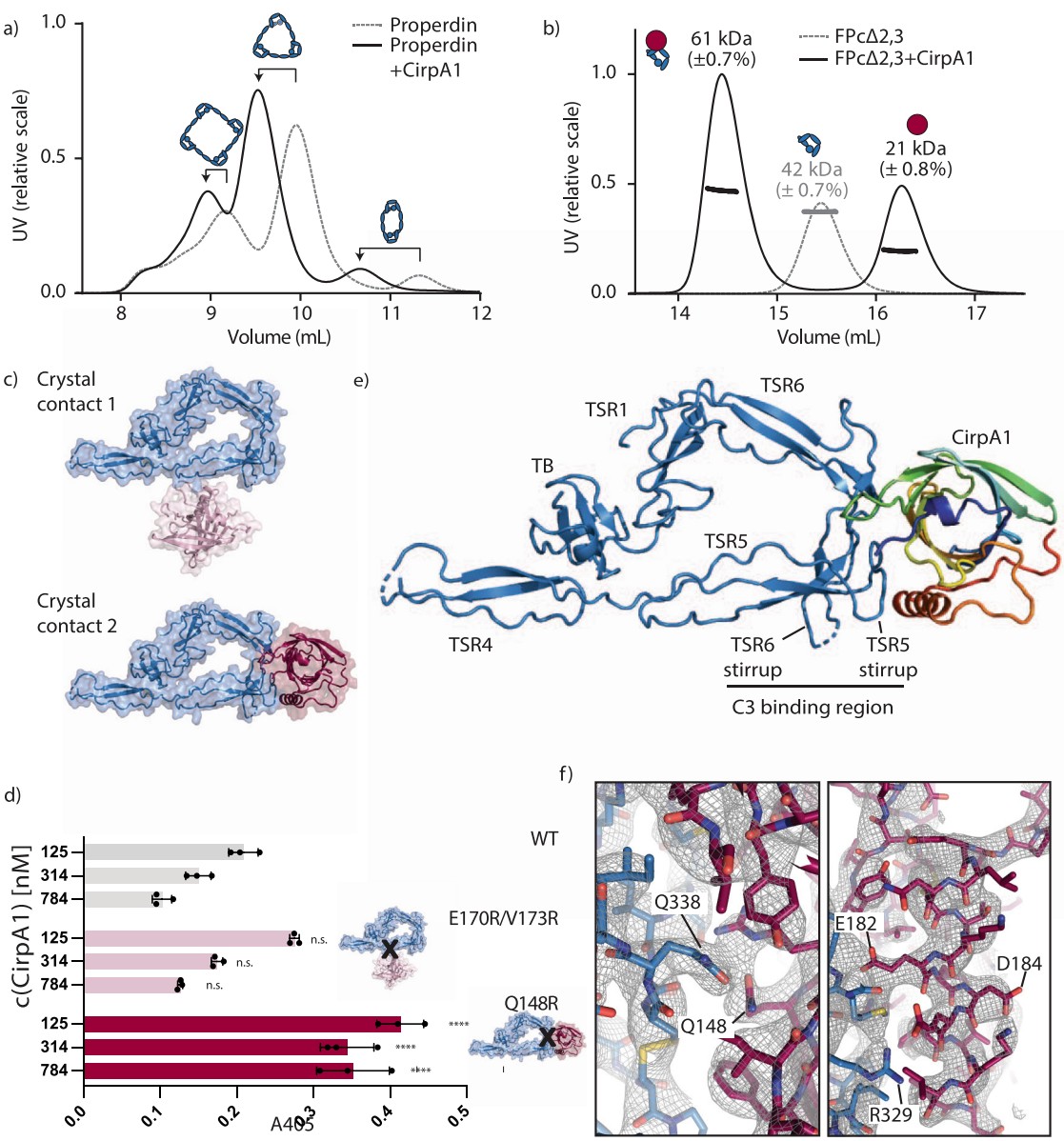

**Fig. 3 Analysis of the CirpA1 binding behaviour to human properdin. a** Size exclusion chromatography of properdin isolated from human serum with and without pre-bound CirpA1. **b** SEC-MALS demonstrates binding of CirpA1 to FPΔ2,3 in a 1:1 stoichiometry. **c** Crystal contacts observed between CirpA1 and FPΔ2,3 result in two possible binding sites of CirpA1. **d** AP haemolysis assay with serum from *Homo sapiens* (human), WT = wild-type CirpA1, E170RV173R = CirpA1 carrying point mutation in crystal contact 1, Q148R = CirpA1 carrying point mutation in crystal contact 2. n.s = not significantly different from wild type ($p > 0.001$, **** = significantly different from WT ($p < 0.0001$) as assessed by two-way ANOVA. Errors bars are Mean with s.d. from three technical replicates ($n = 3$). **e** Crystal structure of the complex between FPΔ2,3 (blue) and CirpA1 (rainbow). **f** Close up of the 2Fo-Fc, ac map in the FPΔ2,3-CirpA1 interface region (left) and the CirpA1 C-terminal helix. Source Data are provided as a Source Data file.

isolated FPΔ2,3 and the C3b-c345c-FPΔ2,3 complex, together with a moderate resolution structure of convertase-bound pseudo monomeric properdin, have established that properdin interacts with the convertase mainly through contacts with the C-terminal C345c domain of C3b[18–20]. It binds the convertase via TSR5 and TSR6 near the factor B binding site. It is plausible that binding is strengthened through direct contacts between properdin and Bb, but resolution limitations did not allow us to unambiguously pinpoint interacting residues. Two distinct loops in properdin were shown to play a crucial role in binding to C3b: residues 328–333 (336 in ref. [18]) (TSR5-stirrup or thumb) and residues 419–426 (TSR6-stirrup or index finger). They embrace the C-terminal helix of C3b and might be involved in stabilising the Mg2+ coordination between C3b and Bb.

In context of the properdin bound C3 convertase, binding of CirpA1 results in a steric clash between CirpA1 and the protease component Bb. This direct competition might contribute to the CirpA1 inhibitory effect in vivo. However, it cannot be the only component explaining CirpA1-mediated inhibition as our work shows that CirpA1 prevents properdin binding to C3b in the absence of FB, and promotes dissociation of a preformed C3b-properdin complex. Therefore, the CirpA1 inhibitory mechanism is likely to interfere directly with the properdin-C3b interaction. Notably, the TSR5-stirrup is directly involved in CirpA1 binding. The TSR6-stirrup lies in immediate vicinity to another the CirpA interacting region and appears to be disordered in the CirpA1-FPΔ2,3 structure. While it is similarly flexible in isolated FPΔ2,3 it assumes a stable conformation upon C3b binding. We,

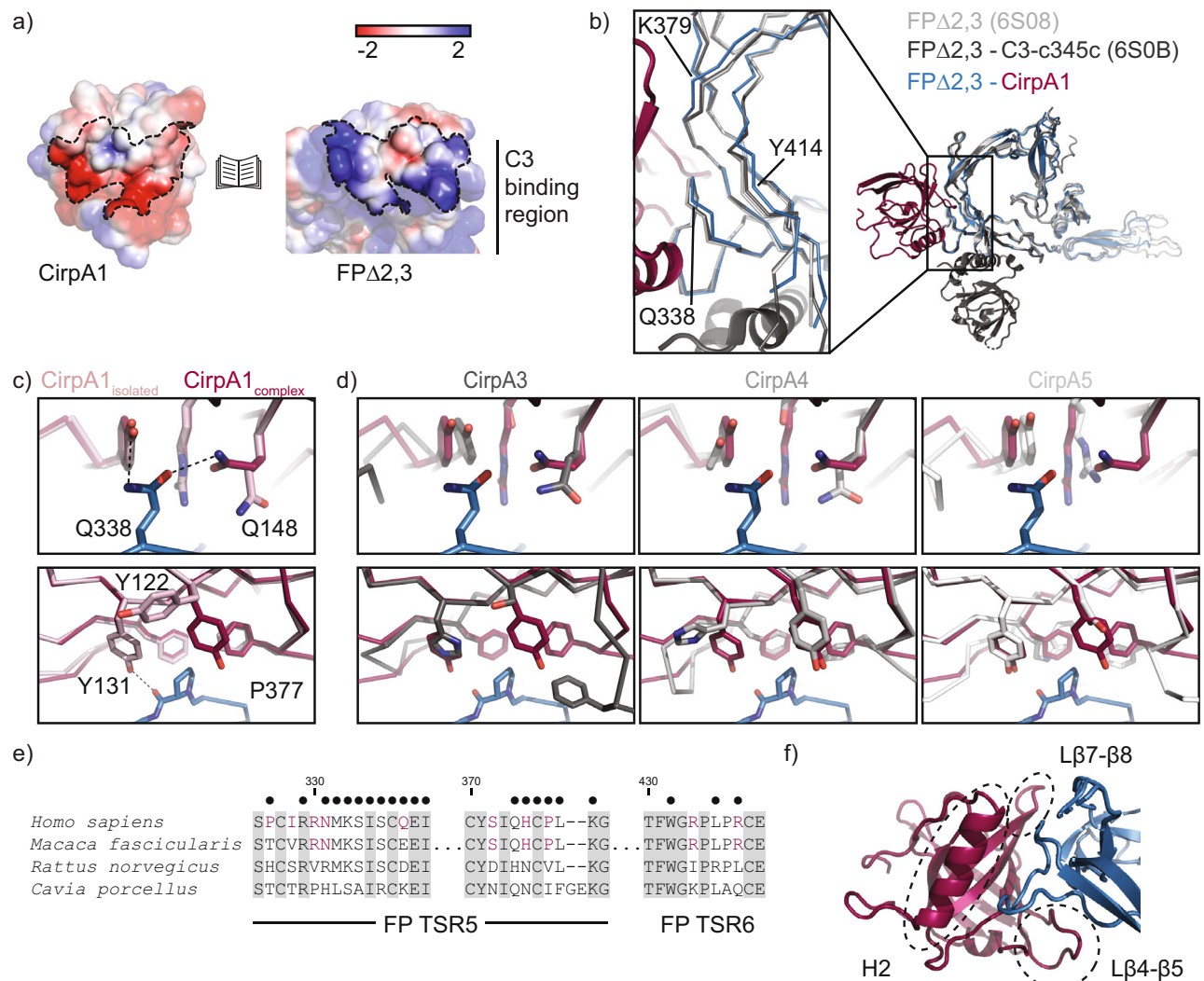

**Fig. 4 Detailed analysis of the CirpA1- properdin interaction. a** Positive (blue) and negative (red) electrostatic potential regions mapped onto the surface representations of CirpA1 and FPΔ2,3 from the FPΔ2,3-CirpA1 complex. Interface region area highlighted by dotted lines. **b** Overlay of structural models of isolated FPΔ2,3 (PDB 6S08, light grey), FPΔ2,3-C3-c345c (PDB 6S0B, dark grey) and FPΔ2,3-CirpA1 (PDB 7B26, blue/magenta) aligned to properdin residues 320–440. Close up of CirpA1 interfacing region. **c** Structural changes in CirpA1 upon complex formation with FPΔ2,3. **d** Superposition of structural models of CirpA3-5 (shades of grey) with the CirpA1- FPΔ2,3 complex, view as in (**c**). **e** Multiple sequence alignment of properdin from species used in AP haemolysis assays in Fig. 2b, showing the regions interacting with CirpA1, red: residues unique to species inhibited by CirpA1, black dots: residues involved in the CirpA1- FPΔ2,3 interface in the complex structure in Fig. 3e according to PISA analysis. **f** Close-up view of the CirpA1- FPΔ2,3 interface, regions of high conformational variability between CirpA homologs as identified in Fig. 2d highlighted with dashed lines.

therefore, rationalise that binding of CirpA1 impacts the conformational freedom of the stirrup loops thereby destabilising the interaction with C3b and possibly preventing conformational changes required for binding to the convertase.

Combining together the results from our structural and functional analyses, we are proposing a model for CirpA1-mediated complement inhibition that mimics the decay acceleration models of convertase regulation (Fig. 5g). In the absence of the inhibitor, the C3 convertase is bound and stabilised by multimeric properdin. In the presence of CirpA, the inhibitor binds free properdin at the TSR5/6 junction, thereby blocking properdin from binding the C3 convertase. Furthermore, CirpA1 is capable of binding to convertase-engaged properdin molecules, triggering properdin displacement.

In summary, our study represents a comprehensive characterisation of a human properdin inhibitor covering isolation from its natural source, target identification, and functional and structural characterisation. In addition, our structure of inhibitor-bound properdin sheds light on the mechanisms involved in properdin-convertase binding, thereby marking an important contribution to understanding the human complement response.

## Methods

**Ethics statement**. Written informed consent for donation of human blood was obtained from healthy donors and all cellular material was destroyed immediately. Use of the remaining serum as a reagent was carried out under the guidance of Oxford University OHS policy document 1/03.

**Fractionation of *R. pulchellus* Salivary Glands**. *R. pulchellus* ticks were reared, and 250 salivary glands from 6-day fed female *R. pulchellus* were dissected according to Tan et al.[27]. The gland protein extract was topped up with 25 mM $Na_2HPO_4/NaH_2PO_4$, pH 7.0 to 10 mL. The sample was then fractionated by sequential anion exchange, hydrophobic interaction chromatography, and size exclusion chromatography (SEC). At each stage, eluted fractions and flow-through from the chromatographic columns were assayed for complement inhibitory activity, and the active fractions were further fractionated. First, protein extract was fractionated by anion exchange chromatography using a MonoQ 5/50 GL column (GE Healthcare), washed with 10 column volumes (CV) 25 mM $Na_2HPO_4/$

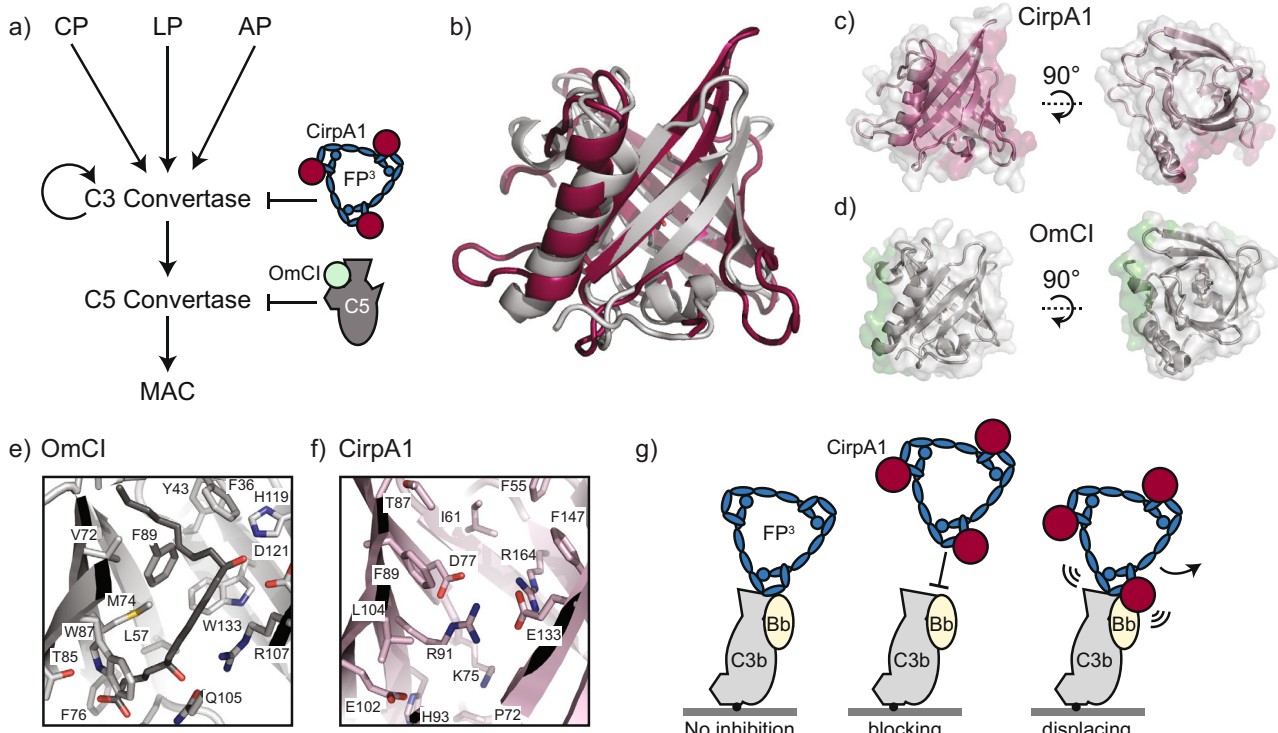

**Fig. 5 Broader context of CirpA1-mediated complement inhibition. a** Schematic representation of the complement cascade indicating the point of inhibition of CirpA1 (dark red) in comparison to OmCI (light green). **b** Overlay of structural models of CirpA1 (dark red) and OmCI (grey). **c** Side and top views of the CirpA1 structure, regions on the CirpA1 surface representation involved in FPΔ2,3 binding are highlighted in red. **d** Side and top views of the OmCI structure, regions on the OmCI surface representation involved in C5 binding are highlighted in green. **e** Close-up of the OmCI ligand binding pocket, bound to leukotriene B4 (PDB 3zuo). **f** CirpA1 region corresponding to the OmCI pocket shown in (**e**). **g** Cartoon representation of the proposed mechanism of CirpA1-mediated complement inhibition via blocking of properdin binding (centre) or displacing pre-bound properdin (right) in contrast to undisturbed convertase binding in absence of CirpA1 (left).

NaH$_2$PO$_4$, pH 7.0, and eluted by a 0–0.5 M NaCl gradient over 30 CV in 500 µL fractions. Active fractions were combined, mixed with an equal volume of 3.4 M (NH$_4$)$_2$SO$_4$, pH 7.0, centrifuged (22,000 × $g$, 10 min), and the supernatant topped up to 5 mL with 1.7 M (NH$_4$)$_2$SO$_4$, 100 mM Na$_2$HPO$_4$/NaH$_2$PO$_4$, pH 7.0. The sample was loaded onto a 1 mL HiTrap Butyl HP column (GE Healthcare) and washed with 5 CV of 1.7 M (NH$_4$)$_2$SO$_4$, 100 mM Na$_2$HPO$_4$/NaH$_2$PO$_4$, pH 7.0. Elution was carried out by a 1.7 to 0.0 M (NH$_4$)$_2$SO$_4$ gradient over 15 CV in 1 mL fractions. The active fraction was topped up to 220 µL with PBS, separated by a Superdex S75 10/300 column (GE), and collected in 250 µL fractions. All fractions were buffer exchanged to PBS and concentrated.

**Identification of CirpA1**. Identified protein fractions with complement-inhibitory abilities were digested by Trypsin and analysed by LC-MS/MS. Samples were topped up to 50 µL with 50 mM TEAB, pH 8.5, reduced with 20 mM TCEP (21 °C, 30 min), alkylated with 50 mM chloroacetamide in the dark (21 °C, 30 min), digested with 0.5 µg of trypsin (37 °C, 16 h), then quenched with 1 µL formic acid. Digested peptides were analysed by LC-MS/MS over a 30-min gradient using LTQ XL-Orbitrap (Thermo Scientific) at the CentralProteomicsFacility (http://www.proteomics.ox.ac.uk, Sir William Dunn School of Pathology, Oxford, United Kingdom). Data were analysed using the central proteomics facilities pipeline[31] and peptides were identified by searching against two *R. pulchellus* sialome cDNA databases[26,27] with Mascot (Matrix Science). Hits were assessed for the presence of a signal peptide with the SignalP 4.1 Server[32] (Copenhagen Business School, Technical University of Denmark), sequence homology to known protein sequences by blastp (NCBI), and structural homology to known protein structures by FFAS[33].

**Expression and purification of recombinant proteins**. Codon-optimised Gen-eArt (ThermoFisher Scientific) strings were cloned into pExpreS2-2 vector with the insect BiP signal sequence followed by an N-terminal 6-His tag. Transfections into *Drosophila melanogaster* S2 suspension cells were carried out following the manufacturer's instructions (Expres2ion Biotechnologies).

CirpA1 was codon-optimised, cloned into pETM-14 using the *Nco*I and *Not*I restriction enzymes, and transformed into *E. coli* BL21 (DE3) (New England Biolabs). CirpA3, CirpA4, and CirpA5 were codon optimised, cloned into pET15b

vector using the restriction enzymes *Nco*I and *Xho*I and transformed into *E. coli* BL21 (DE3) (New England Biolabs). Protein expression was carried out in LB broth (with 50 µg/mL kanamycin). Cells were induced with 1 mM isopropyl-β-D-thiogalactopyranoside (IPTG) for 16 h at 20 °C, resulting in expression in inclusion bodies.

The supernatant was harvested and filtered. Expressed proteins were subsequently purified using a cOmplete His-Tag Purification column (1 mL column, Roche) and SEC (S200, 16/60, GE Healthcare) in PBS.

Cells were harvested by centrifugation and lysed by homogenisation. The lysate was spun at 30,000 × $g$, 4 °C for 30 min and the cell pellet fraction was re-suspended in 40 ml of Buffer A (50 mM Tris, 150 mM NaCl, pH 8, 0.05% (v/v) Tween 20) using a hand-held homogeniser. After incubation on a rotary wheel at 4 °C for 1 h, the re-suspended pellet was centrifuged at 30,000 × $g$ at 4 °C for 30 min. This wash step was repeated once more. The cell pellet was then resuspended in 40 ml of Buffer B (8 M urea, 1 mM EDTA, 100 mM Tris, 25 mM DTT, pH 8) for solubilisation of inclusion bodies, and incubated on a rotary wheel at 4 °C for 2 h. Following centrifugation, the supernatant fraction was retained and filtered. For refolding, the supernatant was added dropwise to Buffer C (1 mM cysteine, 2 mM cystine, 20 mM ethanolamine, 1 mM EDTA, pH 11) with stirring. The protein-containing refolding buffer was left overnight at 4 °C, and then concentrated down to ~40 ml using a 10 kDa Vivaflow 200 filtration device (Sartorius). For CirpA1, the His-Tag was removed following concentration via cleavage with 3C protease whilst dialysing against 2 L of Buffer D (50 mM Tris, 150 mM NaCl, pH 8) overnight at 4 °C. Uncleaved material was removed by reverse nickel purification. All inhibitors were purified by SEC (S75, 26/60, GE Healthcare) in Buffer D. CirpA3, CirpA4, and CirpA5 were further purified by ion exchange chromatography (Mono Q 5/50 column, GE Healthcare). Homogeneous protein populations eluted in the peak (CirpA3, CirpA4) and flow-through (CirpA5) fractions and were dialysed against 50 mM Tris, 20 mM NaCl, pH 8.5, overnight at 4 °C.

**Purification of human properdin from serum**. An affinity column was generated with CirpA1 using the Pierce NHS-Activated Agarose Slurry (Thermo Scientific) following the manufacturer's instructions. In brief, CirpA1 was mixed with the slurry on a rotary wheel overnight at 4 °C. After coupling, the remaining active sites

were blocked with 1 M ethanolamine. The slurry was then packed into an empty cartridge to generate a CirpA1 'column'. Outdated normal human serum was acquired from the John Radcliffe Hospital, Oxford. After loading the serum onto the CirpA1 column and washing with PBS, properdin was eluted with 0.2 M gly-cine-HCl, pH 3. The pH of the elution fraction was adjusted by adding 50 μl of neutralisation buffer (1 M Tris, pH 9) per 1 ml of eluate. Elution fractions were further purified by SEC (S200 10/30 GE Healthcare) in PBS, pH 7.4. The presence of properdin in the elution fraction was confirmed by SDS-PAGE and Western blot.

**Expression and purification of human monomeric properdin lacking TSR 2, 3.** Monomeric properdin lacking TSR domains 2 and 3 was prepared as previously described[18].

Purified FPΔ2,3 was mixed with refolded CirpA1 at a 1:1.5 ratio and incubated for 5 min at RT. The complex was separated from excess CirpA1 by SEC (S200 16/60 GE Healthcare) in 20 mM Tris/HCl pH 7.5, 100 mM NaCl. A 1:1 stoichiometry was confirmed using SEC-MALS (S200, 10/300 GL GE Healthcare).

**Complement inhibition assays.** Complement inhibition ELISAs were performed using a Wieslab complement system screen (Euro Diagnostica) following the manufacturer's instructions, with sample added prior to serum. Assays for the classical and lectin pathways were performed with sheep red blood cells (TCS Biosciences) sensitised with 1:2000 anti-sheep red blood cell stroma antibody (cat. no. S1389, Sigma-Aldrich), alternative pathway assay was performed with rabbit red blood cells (TCS Biosciences). Human serum dilution of 1:18, 1:101 and 1:101 were used for the AP, CP and LP, respectively. Alternative pathway haemolysis inhibition assays were carried out with rabbit red blood cells as described previously[28]. Briefly, 5 ml of rabbit erythrocytes in Alsever's solution (TCS Bios-ciences) were mixed with 20 ml of AP haemolysis buffer (0.1 M HEPES, 0.15 M NaCl, 8 mM EGTA, 5 mM MgCl$_2$, 0.1% (w/v) gelatin, pH 7.4), and centrifuged for 5 min at 1250 × g. Cells were washed in AP buffer until supernatant was clear and cells were resuspended in 15 ml of AP buffer for assays. Serial dilutions of tick inhibitors purified from insect cells were used to assay interspecies activity whereas wt and mutant CirpA1 purified from E.coli was used to distinguish Properdin binding sites. Briefly, fifty microliters of cells (2 × 10$^8$ cells/mL) were incubated in an equal volume of diluted serum (1 h, 37 °C, shaking), supplemented with 2 μL of purified inhibitor or control. Cells were pelleted and haemolysis was quantified at A405 nm of supernatant. Cells were serum only were used for normalisation (100% activity). Final serum dilutions used was as follows: 1:5 (human), 1:6 (monkey), 1:3 (rat) and 1:5 (guinea pig). Human serum from healthy volunteers was prepared as previously described[28]; Macaca fascicularis serum was a kind gift from John Davis and Elena di Daniel; rat and guinea pig serum were from Complement Technology Inc (CompTech, USA).

**Pull-down assay.** For pull-down assay, 0.1 mg/mL of purified protein was immobilised on Pierce NHS-activated magnetic beads (Thermo Fisher) following the manufacturers' instructions. The beads were incubated with 10 mM EDTA and 50 μL serum (21 °C, 30 min). The beads were washed 3 times with 1 mL PBS + 0.05% (v/v) Tween20, once with 100 μL PBS, and boiled in 50 μL SDS-PAGE loading buffer. Elutions were analysed by SDS-PAGE and semi-dry Western blotting.

**Competition assay.** Biotinylated C3b (bC3b) was prepared from C3 (CompTech, USA) as previously described[4]. Briefly, purified C3 was incubated with 1 μg/ml trypsin in the presence of 100 μg/ml maleimide-PEG2-biotin (Thermo Scientific, USA) for 10 min at 37 °C, followed by inactivation with soybean trypsin inhibitor and iodoacetamide. 0.1 mg/mL bC3b in PBS was immobilised on MagStrep "type3" XT Beads (iba) using 250 μL resin per mL. To prevent complement activation and thereby uncouple the effects of properdin binding to C3b vs. properdin binding to fully assembled convertases, human serum was pre-treated with 10 mM EDTA and 10 mM EGTA. Pre-inhibited serum was prepared by adding 10 μM CirpA1 to the inactivated serum and incubating 30 min at 25 °C. To investigate Properdin binding to C3b in the presence or absence of CirpA1 25 uL bC3b bound beads were incubated for 2.5 h at 25 °C in 100 μL of inactivated serum or 100 μL pre-inhibited serum. 25 μL beads without bC3b were incubated with inactivated serum as a negative control. The beads were washed 5 times with 100 μL PBS. To study competitive binding of CirpA1, 25 μL of beads that had not previously contained CirpA1 were subsequently incubated for 30 min with 100 μL 10 μM CirpA1 in PBS at 25 °C. The FT was collected, and the beads were washed five times with 100 μL PBS. Beads from all reactions were resuspended in twice the bead volume of SDS sample buffer and heated to 95 °C for 10 min to elute all bound protein immedi-ately after the last wash. The elutions from all reactions as well as the flow-through from the competition sample were analysed by SDS-PAGE followed by semi-dry Western Blotting against Properdin.

**Semi-dry western blotting.** For blotting the SDS/PAGE-separated proteins were transferred to a PVDF membrane (Amersham Hybond P0.2 PVDF, 55 GE) by semiwet transfer (Bio-Rad) and blocked for 1 h with PBS/2% milk.

Primary antibodies were purchased from CompTech (Complement Technologies Inc, USA): goat α-properdin, 1:2,000, goat α-Factor B, 1:4000, goat α-Factor D, 1:250. Secondary antibody (donkey α-goat HRP, Promega, 1:10,000). For His-tagged proteins, the Penta-His HRP Conjugate Kit (Qiagen) was used following the manufacturer's instructions.

Blots were developed using ECL Western Blotting Substrate (Promega) and imaged using Amersham Hyperfilm ECL (GE Healthcare) or using a ChemiDoc XRS + imaging system (Biorad).

**SEC-MALS.** For SEC-MALS, 100 μL of protein sample at 1 mg/mL was injected onto an S200 10/300 column (GE Healthcare) equilibrated in PBS. Light scattering and refractive index were measured using a Dawn Heleos-II light scattering detector and an Optilab-TrEX refractive index monitor. Analysis was carried out using ASTRA 6.1.1.17 software assuming a dn/dc value of 0.186 mL/g.

**Microscale thermophoresis.** Monomerized FPΔ2,3 was labelled using the protein labelling kit RED-NHS (NanoTemper Technologies) following the manufacturer's instructions but increasing the incubation time to 2 h. Labelled FPΔ2,3 was sepa-rated from unlabelled protein by SEC using a Superdex200 5/150 column in PBS, 0.05% (v/v) Tween-20, ph7.4. In the MST experiment, labelled FPΔ2,3 was kept at a constant concentration (50 nM), while the concentration of the CirpA1 was varied between 2 nM and 64 μM. After 5 min incubation, the samples were loaded into Monolith NT.115 capillaries (NanoTemper Technologies) and the MST measure-ments were performed using the Monolith NT.115 at 20% LED power and low MST power. An MST on time of 2.5 s was used for analysis via the MO.Affinity Analysis 2.3 Software.

**Circular dichroism.** CD spectra were collected using a Jasco J-815 CD spectro-photomer. Proteins were dialysed against 10 mM Na$_2$HPO$_4$/NaH$_2$PO$_4$, pH 7.4 and diluted to 0.1–0.2 mg/ml (150 μL). Experiments were performed at 20 °C using a cuvette of 1 mm path length (Starna Scientific). CD spectra were collected in the wavelength range of 190–260 nm with four accumulations.

**Crystallisation, X-ray data collection, and structure determination.** Refolded CirpA1 in 50 mM Tris, 150 mM NaCl, pH 8 was concentrated to 17.8 mg/mL. The protein was mixed with an equal volume of mother liquor containing 0.2 M imi-dazole malate, pH 6, 30% (w/v) PEG4000, and crystallised in 300 nL drops by a vapour-diffusion method at 21 °C. Crystals were cryoprotected in mother liquor supplemented with 20% (v/v) PEG400 and flash-frozen in liquid N$_2$. Data were collected on beamline I04 at the Diamond Light Source (Harwell, United King-dom), wavelength: 0.9795 Å, as specified in Table 1. The structure of CirpA1 was solved by molecular replacement using MolRep[34] within CCP4[35]. The search model was created using Chainsaw in CCP4[36] based on the structure of OmCI (PDB ID code 3ZUI). The initial model of CirpA1was built and refined through iterations of Buccaneer[37] and REFMAC5[38]. Subsequently, the model was subjected to multiple rounds of manual rebuilding in Coot[39] and refinement in REFMAC5[38] or Phenix[40].

Initial crystallisation trays were set up with full length refolded CirpA3. Crystal quality could be substantially improved by using a truncated CirpA3 construct (encompassing residues 6–160). Refolded, truncated CirpA3 (6–160) in 50 mM Tris, 150 mM NaCl, pH 8 was concentrated to 13.1 mg/mL. The protein was mixed in a 1:3 ratio with mother liquor containing 0.2 M ammonium sulphate, 30% (w/v) PEG 4000, and crystallised in 300 nL drops by a vapour-diffusion method at 21 °C. Crystals were cryoprotected in mother liquor supplemented with 20% (v/v) PEG400 and flash-frozen in liquid N$_2$. Data were collected on beamline I04-1 at the Diamond Light Source (Harwell, United Kingdom), wavelength: 0.9762 Å, as specified in Table 1. The structure of CirpA3 was solved by molecular replacement using MolRep[34] within CCP4[35]. The search model was created using Chainsaw in CCP4[36] based on the structure of CirpA1. Subsequently, the model was subjected to multiple rounds of manual rebuilding in Coot[39] and refinement in Phenix[40]. The presence of twinning was identified by Xtriage[41] based on the intensity statistics. Therefore, the final stages of refinement in PHENIX were performed using the twin law (l,−k,h) as obtained from Xtriage.

Refolded CirpA4 in 50 mM Tris, 150 mM NaCl, pH 8 was concentrated to 9.6 mg/mL. The protein was mixed in a 1:3 ratio with mother liquor containing 0.005 M cadmium chloride hemi(pentahydrate), 0.1 M sodium acetate, pH 5.5, 20% (w/v) PEG 4000, and crystallised in 300 nL drops by a vapour-diffusion method at 21 °C. Crystals were cryoprotected in mother liquor supplemented with 20% (v/v) PEG400 and flash-frozen in liquid N$_2$. Data were collected at the MASSIF1 beamline of ESRF by automatic data collection, using an X-ray beam of wavelength 0.966 Å, as specified in Table 1. The structure of CirpA4 was solved by molecular replacement using MolRep[34] within CCP4[35] with the structure of CirpA1. Subsequently, the model was subjected to multiple rounds of manual rebuilding in Coot[39] and refinement in Phenix[40].

Refolded CirpA5 in 50 mM Tris, 150 mM NaCl, pH 8 was concentrated to 21.5 mg/mL. The protein was mixed in a 1:3 ratio with mother liquor containing 0.1 M Sodium HEPES, pH 8.2, 50% (v/v) PEG500 MME, and crystallised in 300 nL drops by a vapour-diffusion method at 21 °C. Crystals were cryoprotected in mother liquor and flash-frozen in liquid N$_2$. Data were collected on beamline I04-1

at the Diamond Light Source (Harwell, United Kingdom), wavelength: 0.9159 Å, as specified in Table 1. The structure of CirpA5 was solved by molecular replacement using MolRep[34] within CCP4[35] using the structure of CirpA1 as a search model. The structure of Cirp-A5 was built and refined through cycles of automated model building by Buccaneer[37] and refinement by REFMAC5[38]. Subsequently, the model was subjected to multiple rounds of manual rebuilding in Coot[39] and refinement in REFMAC5[38] or Phenix[40].

CirpA1 was copurified with the FPΔ2,3 by SEC (S200 16/60 GE Healthcare) in 20 mM Tris/HCl pH 7.5, 100 mM NaCl and concentrated to 10.5 mg/mL. The protein was mixed with an equal volume of mother liquor containing 0.15 M Potassium thiocyanate, 0.1 M Tris, pH 7.5, 18 % w/v PEG 5000 MME, and crystallised in 200 nL drops by a vapour-diffusion method at 21 °C. Crystals were cryoprotected in mother liquor supplemented with 20% (v/v) PEG400 and flash-frozen in liquid N2. Data were collected on beamline I04-1 at the Diamond Light Source (Harwell, United Kingdom), wavelength: 0.915890 Å, as specified in Table 1. The structure of the complex was solved by molecular replacement using MolRep[34] within CCP4 with the structures of CirpA1 (PDB ID code 7BD2, this study) and FPΔ2,3 (PDB ID code 6S08). The initial model was subjected to multiple rounds of manual rebuilding in Coot[39] and refinement in Phenix[40]. The protein chemistry of the final models was validated using MolProbity[42]. The structures are characterised by the statistics shown in Table 1. Interactions between CirpA1 and properdin have been analysed by PDBePISA[43]. Protein structure figures were prepared using Pymol v2.4 (Schrödinger). Electrostatic potentials were calculated using the APBS programme in Pymol[44].

**Reporting summary**. Further information on research design is available in the Nature Research Reporting Summary linked to this article.

## Data availability

X-ray coordinates and data have been deposited into the Protein Data Bank (PDB), www.pdb.org under accession codes PDB 7B2D (CirpA1), PDB 7B28 (CirpA3), PDB 7B29 (CirpA4), PDB 7B2A (CirpA5), PDB 7B26 (CirpA1-FPΔTSR2,3). CirpA inhibitor sequences have been deposited into the GenBank under accession codes MW260265 (CirpA1), MW260267 (CirpA2), and MW260266 (CirpA3); or have already been publicly available under accession codes CD794868 (CirpA4), GEFJ01011401 (CirpA5) and CK182034 (CirpA6). Source data are provided with this paper.

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

## Acknowledgements

We acknowledge the Diamond Light Source and the staff of beamlines I04 and I04-1 for access under proposals mx12346 and mx18069. We further acknowledge the European Synchrotron Radiation Facility and the staff of beamline MASSIF1. We thank M. Slovak (Institute of Zoology, Bratislava, Slovakia) for providing salivary glands; John Davis and Elena di Daniel (Oxford Drug Discovery Institute) for providing *Macaca fascicularis* serum; Simon Newstead (Biochemistry, Oxford) for assistance with Synchrotron data collection. This work was financially supported by the European Molecular Biology Organisation long-term postdoctoral fellowship ALTF 554-2019 to K.B.; Wellcome Trust grants 100298, 209194, and 219477 to S.M.L; and Medical Research Council UK grant S021264 to S.M.L. This research was supported [in part] by the Intramural Research Program of the NIH.

## Author contributions

M.M.J., K.B., J.A., S.J., G.R.A., and S.M.L. designed research; K.B., J.A., M.M.J., S.J., T.T., D.V.P., and S.M.L. performed research; K.B. J.A., S.J., M.M.J., and S.M.L. analysed data; and K.B., S.J., and S.M.L. wrote the paper.

## Funding

## Competing interests

The authors declare no competing interests.

## Additional information

**Peer review information** *Nature Communications* thanks Bryan Morgan, Claudia Kemper, and the other anonymous reviewer(s) for their contribution to the peer review this work. Peer reviewer reports are available.

