## [Peer Review File · Nature Communications]

Reviewers' Comments:

Reviewer #1:

Remarks to the Author:

This manuscript describes the discovery of novel inhibitor of the complement pathway, CirpA, that targets properdin. There is a description of the discovery from the saliva of a tick, identification of the complement pathway affected and finishing with the structural characterisation of the binding interface between CirpA and properdin. There is some validation of the inhibitor binding site. Overall this is a very promising study of high impact however it is still very much in a preliminary state and needs more thorough experimentation to be performed. Experiments have been suggested to help this study become indisputable and of high value to the community.

MAJOR COMMENTS

Page 2 lines 22-24: Regarding maintaining CP and LP pathways while inhibiting AP. I feel that after reading the paper that this can't really be concluded or even introduced as a concept because there is no proof that a FP inhibitor doesn't affect these pathways at higher concentration. Fig 1c only deals with μM concentrations and the assay is cut-off where there is potential indicative effect on the CP at high concentrations of CirpA. The AP is really just the amplification step for all pathways so it is really not that clear cut with respect to inhibition of pathways. In general, this comment is really out of place and potentially deceiving for non-complement experts.

Page 3 line 15: this should really state that CirpA inhibits HUMAN complement activation. The other proteins might inhibit the complement pathways of the common hosts of the mite (typically ungulate species).

Page 3 line 19: This is the incorrect interpretation of the result. A single hit can not be determined by an antibody that is specific as it doesn't detect other proteins/chemicals in the elution from the CirpA column. The authors should show evidence that there is only one protein that binds to the CirpA column and that this corresponds to FP (Coomassie SDS-PAGE, mass spec analysis)

Page 4 figure 1C: based on the concentration of the protein in crystal trials, you expect to be able to get the protein to at least two orders of magnitude higher in concentration which would be of immense value to the interpretation of the assay. i.e. a concentration of hundreds of micromolar could be tested in these assays. This is important because often you see residual activity and this gives a more quantitative assessment of the activity. I.e. inhibition is not a binary event where it either binds or doesn't. This approach would be of interest to show what happens with the classical pathway as there is a slight indication that there is also inhibition in this pathway (Fig 1c). And this would actually make a lot of sense in complement biology because properdin still contributes to the stabilisation of C3 convertase in the application step. Not just in the initiation of the alternative pathway.

Page 5 line 26: what is the rationale for the use of the complement inhibitor structure of OMCL as the initial search model? There is no rationale presented in either the introduction or in the results as to the knowledge of CirpA being part of the lipocalin fold family. This information seemed to come out of the blue that there was sufficient sequence and structural similarity with OMCL to warrant this being a search model. To improve on this aspect OMCL should be included in the sequence alignment in figure 2 and also a statement of the sequence identity/similarity. Furthermore, one of the latter discussion points is a statement (Page 10 line 21) that the structures were all shown to have lipocalin folds. They were solved by MR so of course the fold will be known.

***Page 7 Lines 20-25: with respect to the mutants, particularly Q148 are mutant, these are very preliminary Studies of the interface. As per figure one and figure 2c, there is capacity to test higher concentrations and therefore see reduced inhibitory activity from the Q148R mutant. There's indications that there is slight inhibitory activity in the graphs in figure 3D. Furthermore, it would be good to have more data points of higher concentration protein in the assays especially when considering the Q148R mutation relative to the charged services of the proposed binding site as shown in figure 4a. It seems unlikely that mutation of a glutamine to an arginine could completely disrupt this interface given the overall charged patches. And again this might be

reflected in the concentration of Inhibition being shifted rather than the impression that it is totally abolished. I.e. it was not tested high enough concentrations to exclude that is merely reduced inhibitory activity rather than no inhibitory activity. This could be easily remedied by tweaking the conditions of the assay to have a broader range of concentration points of the inhibitor and would provide a wealth of information.

Another potential explanation for the Q148R mutant is that it is not properly folded or that the point mutation affects thermal stability of the protein. In order to exclude this data would need to be presented to show that the mutant mutation does not affect the fold compared to wildtype. Possibly using CD spec, thermal melt, SEC, DLS, nano-DFS??

MINOR COMMENTS

Page 1 line 40: For those readers who are not expert in the complement pathway, it would be important to be very accurate with the introduction of the complement pathway and describe how the density leads to C5 convertase and that this is not really just a change in density but a change in protein complex assembly.

Page 2, line 3: It would be good to mention some examples or range in types of inhibitors rather than cite that there are many. This is a really interesting area of research that needs to be promoted.

Page 2, line 5: It is really important to mention which parts of the complement pathway are disproportionate in covid responses (is this too much C3a and C5a leading to inflammation?)

Page 2 line 13: suggest reference to a schematic of the properdin domains in the supplementary data including how the higher order oligomers are arranged.

Page 2 line 40: avoid shortened paragraphs. This was found throughout the manuscript.

Page 4 figure 1F and 1G: why do the molecular weight of prepared and differ in these two Westons?

Page 4 figure 1G: It appears that this western is quite modified with respect to the signal and background of the Western Especially when compared to figure 1f. Could the original Western please be included in the manuscript?

Page 5 line 6 and Page 5 line 21: it's interesting that the different inhibitors were originally expressed using S2 cells, and then later expressed in E. coli cells. Why were proteins originally expressed in S2 cells? Has there been a comparison of the inhibitory activity of both sources of CirpA1 and homologues to ensure QA?

Figure 5 line 9: Considering that these ticks that are studied typically feed on hosts that are either ungulates or bovine species, what is the rationale for only testing human, monkey, rat and guinea pig serum? It would be extremely beneficial to this study in general to routinely test horse and bovine serum which is easily accessible. There may be some important veterinary applications that open up as part of this study.

Page 6 figure 2a: Suggestion: add OMCL to the sequence alignment or put in supplementary data

Page 7 line 17: the statement in the sentence says that the resolution is 3.4 Å, however, the table states a higher resolution. 2.84 Å. The 3.4 Å resolution makes sense when looking at the electron density maps (although it's hard to tell with the resolution of the figures). The choice of high resolution bins is also questionable given the low $i/\sigma I$ values.

Page 8 figure 3C: it is hard to see the relative orientation of CirpA1 on the surface of the truncated prepared in.

Page 8 figures 3D: why is the y-axis using absorbance units rather than percent lysis as shown in

the assays in figure 1 and figure t2? It would be really beneficial for this manuscript if the same assay parameters represented in all of the figures, particularly because these assays are comparing E. coli expressed protein compared to the previous studies which uses S2 expressed material. It would be very interesting to know how well they compare.

Page 7, last paragraph: in order to demonstrate further the importance of the interface between properdin and CirpA1, it would be really good to also test the residues that comprise the properdin binding interface using point mutations in similar to what was performed on CirpA1.

Page 9, line 25: there's a comment that there is high structural similarity between OMCL and CirpA1. This should be extended to the sequence identity the sequence identity

Page 10 figure 5 a: this is a little bit confusing to the non-complement expert reader because in figure 1 you go to show that the inhibitor only inhibits the alternative pathway but in this diagram you have CirpA1 depicted as inhibiting C3 convertase only at the amplification step. Diagram implies that there is no inhibition of the upstream initiation pathways.

Page 10 line 26: the authors do allude to the tech species using cattle. As hosts they should be expanded upon in the introduction and should be incorporated into the lysis essays in figure 2 for a complete story.

Page 11 second paragraph: Have the wt CirpA1, and mutants of CirpA1, been tested on the wild type properdin and imaged using the ultrastructure such as using Cryo EM to correlate binding sites on properdin? How feasible would this be. It would really make the research much structure.

Page 11 line 22: the production of C3b was not described in the methods. How was C3b made?

Methods: the writing of the message looks rather rushed and needs to be fixed up with respect to scientific nomenclature such as the use of subscripts on chemicals italics for species, the source of important things such as antibodies and C3b.

Extended data: there are no legends for the two figures that comprise the extended data.

First first extended data figure: part C: this table doesn't really make much sense What's the point of having a blast searches if it was already known that OmCl was a homologue. Is OmCl in this table?

Reviewer #3:

Remarks to the Author:

Review of the manuscript 'Targeting properdin - Structure and function of a novel family of tick-derived complement inhibitors' by Dr. Susan lea and colleagues.

In this work, the authors identified and functionally characterized a new family of a complement-regulator from the saliva of the tick *Rhipicephalus pulchellus*.

Complement biology is experiencing a bout of renewed interest. Not only because it is becoming increasingly clear that complement dysfunction contributes to a larger range of human diseases than previously thought but also because complement is emerging as a major culprit in the pathogenesis of severe COVID-19.

Early complement-targeting strategies has not been as successful as initially hoped for. This is in part due to the facts that complement is overall more complex in its effects on other immune sensors/effector systems than previously thought, that it also exerts (immune) homeostatic and tissue repair functions, and is operative at unexpected locations well beyond the vascular space. Thus, new tools allowing to uniquely modify this system will be helpful. Properdin is the only known positive regulator of complement. Properdin stabilizes the C3 convertase and thus supports prolonged and more effective alternative pathway activation. There is thus clinical interest to target properdin and several blocking anti-properdin Abs have been developed.

Here, the group of Dr. Lea has identified the first 'natural' properdin inhibitor from a tick species. This protein, CirpA1, a member of the lipocalin family proteins, binds properdin and leads to potent inhibition of AP activity. The group further identify CirpA homologues from other tick species, present crystal structures of these (in part with essential properdin domains) and use these to elegantly demonstrate why the different CirpA family members display AP control host serum species specificity.

The work is important and timely and all experiments are solidly performed with data interpretations accurate.

Experimentally, I could only argue that a set of Biacore experiments using purified components of AP C3 convertase formation and CirpA1 would nicely complement the data presented in Figure 1f (and exclude any serum-derived unknown component that may have impact).

Few minor comments:

1. Although mentioned in the title, the abstract should also contain the information that CirpA1 was identified in the tick.
2. The labeling of the figures (particularly Figure 1 with ve- and ve+ denoting different conditions in different sub panels; OmCl is mentioned in the text but not legend, etc.) is really confusing and could be more intuitive. I would suggest to make generally label panels better (for example, Figure 2a could have the tick species connected to the CirpA proteins, etc. etc.).
3. Although data seem to be largely statistically significant 'by eye' (where statistical analyses are applicable), they should still be included (Fig 1c, 2b, 3d, etc.).
4. According to the complement nomenclature update 2019, properdin should be called 'properdin' and not FP.
5. The authors claim that their work represents the discovery of the first properdin inhibitor. It may be more accurate to mention that anti-properdin antibodies exist and work but that they identified the first 'natural' inhibitor.

Reviewer #4:

Remarks to the Author:

This is an interesting and generally well-written manuscript describing the isolation from ticks of a protein inhibitor of properdin; binding of the tick protein prevents or reverses properdin binding to the amplification loop convertase and thus reduces convertase stability and complement activation downstream. Given the current huge interest in complement inhibiting drugs the work is of broad potential interest. The approach is not completely novel in that there are already several publications describing mAb against properdin that work in a similar manner and indeed these are progressing towards the clinic. Nevertheless, the thorough characterisation of a nature-derived inhibitor is important.

I have a few comments that I hope will improve clarity of the work:

1. A terminology issue (sorry!) - Factor P (FP) is not accepted terminology for properdin; the 2014 nomenclature report from ICS and ECN did not recommend change and even the 2019 opinion piece from Tenner and others (Front Immunol.2019 Jun 7;10:1308) that has gained traction in the field does not recommend changing the name. I am aware that others have used FP recently but given the likely influence of this article I really urge that the correct and approved nomenclature is used.
2. The statement in the introduction implies that the CP and LP directly lead to formation of the C3bBb convertase - although I know that this was not intended by the authors, the language is confusing - so just stress that this is the AP/amplification loop convertase.
3. Figure 1c; using the Weislab assay there is clear inhibition of the AP but NO inhibition of the CP/LP - given that the AP loop is the key amplifier for all pathways this is surprising and likely reflects an artefact of the assay. I would urge repeating using haemolysis assays for at least AP and CP.
4. Figure 1d/e measures C3a/C5a in the Weislab assay supernatants. The data would be more convincing if done outside of the assay with known AP activators - eg. zymosan - enabling dose response analyses.

5. Figure 2b shows the species specificity of the effect - it is surprising given its importance as a model that mouse was not included in this. Also, given later text on natural hosts, why not tested with bovine serum?
6. The crystal structures are very convincing as would be expected from these scientists; however, I don't understand why a truncated properdin is used rather than full-length. Does this not risk missing important interactions? Can the authors explain their reasons?
7. Species specificity is returned to in Discussion - are these ticks known to be primate parasites? It is stated that they have evolved to parasitise cows and other large domestic animals - this deserves attention as suggested above and perhaps by adding in these species to alignments.
8. The discussion around other lipocalins suggests other activities - equally possible that they have no relevant activities or that they do the same job in other species. I'm not sure that the comparison with the C5 inhibitor adds much (at least as currently discussed) and might confuse.
9. The absence of any direct measures of binding affinities using SPR or related methods is surprising - such data could aid understanding of how these proteins can not only prevent properdin binding to convertase but also displace from convertase.
10. Minor annoyances - subscript numbers in chemical names like Na_2PO_4 ; include degree symbol in temperatures like 25°C etc.

B.Paul Morgan

REVIEWER COMMENTS

Reviewer #1 (Remarks to the Author):

This manuscript describes the discovery of novel inhibitor of the complement pathway, CirpA, that targets properdin. There is a description of the discovery from the saliva of a tick, identification of the complement pathway affected and finishing with the structural characterisation of the binding interface between CirpA and properdin. There is some validation of the inhibitor binding site. Overall this is a very promising study of high impact however it is still very much in a preliminary state and needs more thorough experimentation to be performed. Experiments have been suggested to help this study become indisputable and of high value to the community.

MAJOR COMMENTS

Page 2 lines 22-24: Regarding maintaining CP and LP pathways while inhibiting AP. I feel that after reading the paper that this can't really be concluded or even introduced as a concept because there is no proof that a FP inhibitor doesn't affect these pathways at higher concentration. Fig 1c only deals with uM concentrations and the assay is cut-off where there is potential indicative effect on the CP at high concentrations of CirpA. The AP is really just the amplification step for all pathways so it is really not that clear cut with respect to inhibition of pathways. In general, this comment is really out of place and potentially deceiving for non-complement experts.

We apologise for our lack of clarity in this paragraph and have rewritten the text. We agree with the reviewer in that inhibition of the AP has the potential to inhibit CP/LP in vivo by way of inhibition of the amplification step. The point we intended to make here is that there is the potential to target the AP pathway specifically by inhibiting Properdin, while leaving the CP and LP arms intact (therefore not completely ablating C5 activation). This is in contrast to the approach of inhibiting the C5 cleavage point that many therapeutic strategies use. Whilst we can't rule out inhibition of CP and LP pathways at high inhibitor concentration, Figure 1c does show that there is a large concentration window in which it is possible to completely inhibit AP without affecting CP and LP. The effect is even more dramatic than Figure 1c suggests, as the CP and LP assays are carried out at a serum (and hence Properdin) concentration that is 5.5 times lower than the AP assay. By our calculations, the inhibitor is in greater than 300-fold excess over Properdin at the highest concentration used in the CP and LP assays in Figure 1c, while the same inhibitor completely inhibits the AP around equimolar concentrations.

Page 3 line 15: this should really state that CirpA inhibits HUMAN complement activation. The other proteins might inhibit the complement pathways of the common hosts of the mite (typically ungulate species).

We apologise and have altered the text to be more specific.

Page 3 line 19: This is the incorrect interpretation of the result. A single hit can not be determined by an antibody that is specific as it doesn't detect other proteins/chemicals in the elution from the CirpA column. The authors should show evidence that there is only one protein

that binds to the CirpA column and that this corresponds to FP (Coomassie SDS-PAGE, mass spec analysis)

We agree with this comment and have altered the text to make clear the fact that multiple proteins were probed for. Given the activity of the inhibitor in specifically knocking down the AP the only possible targets were properdin, Factor B or Factor D. Blots for Factors B & D were negative (new Supplementary Fig. 3). We have also shown that CirpA1 does not bind to purified C3b on agarose beads, and that it is able to both prevent properdin binding to C3b-beads and displaces both properdin and Bb from preformed bead-C3b-Bb-P (Fig 1g).

Page 4 figure 1C: based on the concentration of the protein in crystal trials, you expect to be able to get the protein to at least two orders of magnitude higher in concentration which would be of immense value to the interpretation of the assay. i.e. a concentration of hundreds of micromolar could be tested in these assays. This is important because often you see residual activity and this gives a more quantitative assessment of the activity. I.e. inhibition is not a binary event where it either binds or doesn't. This approach would be of interest to show what happens with the classical pathway as there is a slight indication that there is also inhibition in this pathway (Fig 1c). And this would actually make a lot of sense in complement biology because properdin still contributes to the stabilisation of C3 convertase in the application step. Not just in the initiation of the alternative pathway.

As detailed in the response to major point 1, we have toned down the language of the manuscript regarding specificity of the inhibitor for the AP vs the CP/LP. We believe the important observation is the window of inhibitor concentrations (spanning several orders of magnitude) in which AP inhibition is observed without effect on the CP/LP.

Page 5 line 26: what is the rationale for the use of the complement inhibitor structure of OMCL as the initial search model? There is no rationale presented in either the introduction or in the results as to the knowledge of CirpA being part of the lipocalin fold family. This information seemed to come out of the blue that there was sufficient sequence and structural similarity with OMCL to warrant this being a search model. To improve on this aspect OMCL should be included in the sequence alignment in figure 2 and also a statement of the sequence identity/similarity. Furthermore, one of the latter discussion points is a statement (Page 10 line 21) that the structures were all shown to have lipocalin folds. They were solved by MR so of course the fold will be known.

We apologize and have added a sentence explaining the logic of this search model choice, including the sequence identity. However, we feel that adding the alignment to the figure would serve to confuse, especially given the very low sequence identity (12%), with only the disulphide-bonded cysteines being relevant). We have altered the discussion text to state that the structures confirmed a lipocalin fold.

***Page 7 Lines 20-25: with respect to the mutants, particularly Q148 are mutant, these are very preliminary Studies of the interface. As per figure one and figure 2c, there is capacity to test higher concentrations and therefore see reduced inhibitory activity from the Q148R mutant. There's indications that there is slight inhibitory activity in the graphs in figure 3D. Furthermore,

it would be good to have more data points of higher concentration protein in the assays especially when considering the Q148R mutation relative to the charged services of the proposed binding site as shown in figure 4a. It seems unlikely that mutation of a glutamine to an arginine could completely disrupt this interface given the overall charged patches. And again this might be reflected in the concentration of Inhibition being shifted rather than the impression that it is totally abolished. I.e. it was not tested high enough concentrations to exclude that is merely reduced inhibitory activity rather than no inhibitory activity. This could be easily remedied by tweaking the conditions of the assay to have a broader range of concentration points of the inhibitor and would provide a wealth of information.

We have altered the language to reflect the fact that the Q148R may simply be reduced in activity rather than abolished. However, the goal of this experiment was to distinguish between the two binding interfaces and we believe that this was accomplished with the current data.

Another potential explanation for the Q148R mutant is that it is not properly folded or that the point mutation affects thermal stability of the protein. In order to exclude this data would need to be presented to show that the mutant mutation does not affect the fold compared to wildtype. Possibly using CD spec, thermal melt, SEC, DLS, nano-DFS??

Unfortunately, the data for the mutants was collected just before the UK went into lockdown restrictions in 2020, and we were unable to collect CD data. Subsequently, the lab has moved institute and doesn't currently have the equipment/staff to reproduce the protein samples needed for this experiment.

MINOR COMMENTS

Page 1 line 40: For those readers who are not expert in the complement pathway, it would be important to be very accurate with the introduction of the complement pathway and describe how the density leads to C5 convertase and that this is not really just a change in density but a change in protein complex assembly.

We very deliberately used the language here to avoid the controversy surrounding the existence of distinct C5 convertase complexes. The literature is very clear on the need for additional C3b molecules to shift the specificity of cleavage from C3 to C5. However, there is currently little in the way of clear evidence for the creation of a dedicated C5 convertase enzyme complex, as opposed to the alternative model of the accessory C3b molecules priming the C5 to become a more optimal substrate.

Page 2, line 3: It would be good to mention some examples or range in types of inhibitors rather than cite that there are many. This is a really interesting area of research that needs to be promoted.

We have added some examples of complement regulators.

Page 2, line 5: It is really important to mention which parts of the complement pathway are disproportionate in covid responses (is this too much C3a and C5a leading to inflammation?)

Page 2 line 13: suggest reference to a schematic of the properdin domains in the supplementary data including how the higher order oligomers are arranged.

Added.

Page 2 line 40: avoid shortened paragraphs. This was found throughout the manuscript.

Corrected.

Page 4 figure 1F and 1G: why do the molecular weight of prepared and differ in these two Westons?

We apologize, this was a shift in the labels during figure preparation and has been corrected.

Page 4 figure 1G: It appears that this western is quite modified with respect to the signal and background of the Western Especially when compared to figure 1f. Could the original Western please be included in the manuscript?

The Western in 1g is a crop from the original as recorded by our Bio-Rad Chemidoc system with its automatic exposure software. We have included the uncropped version as part of the re-submission.

Page 5 line 6 and Page 5 line 21: it's interesting that the different inhibitors were originally expressed using S2 cells, and then later expressed in *E. coli* cells. Why were proteins originally expressed in S2 cells? Has there been a comparison of the inhibitory activity of both sources of CirpA1 and homologues to ensure QA?

All potential complement inhibitors were initially expressed in S2 cells in order to maximize the probability of correct folding of disulphide bonds. After CirpA1 activity was identified we shifted to producing it in *E. coli* (with a refold protocol) to make the process of making mutants faster. We did compare the activity of S2 and *E. coli* produced material and have included this as a Supplementary figure, in addition to circular dichroism showing correct folding.

Figure 5 line 9: Considering that these ticks that are studied typically feed on hosts that are either ungulates or bovine species, what is the rationale for only testing human, monkey, rat and guinea pig serum? It would be extremely beneficial to this study in general to routinely test horse and bovine serum which is easily accessible. There may be some important veterinary applications that open up as part of this study.

The focus of the study was identifying inhibitors with potential therapeutic relevance for humans. Hence the focus was initially on primates. We also tested other standard sera routinely kept in the lab for complement inhibition assays, though did not have access to mouse serum at the time these experiments were carried out. The potential importance of ungulates was introduced into the discussion at the time of writing the manuscript. We agree it would be interesting to probe species specificity of the CirpA family further in future work.

Page 6 figure 2a: Suggestion: add OMCL to the sequence alignment or put in supplementary data

As detailed above, we feel that adding the alignment to the figure would serve to confuse, especially given the very low sequence identity (12%), with only the disulphide-bonded cysteines being relevant).

Page 7 line 17: the statement in the sentence says that the resolution is 3.4 Å, however, the table states a higher resolution. 2.84 Å. The 3.4 Å resolution makes sense when looking at the electron density maps (although it's hard to tell with the resolution of the figures). The choice of high resolution bins is also questionable given the low $i/\sigma I$ values.

The table states 2.84 Å for the data processing, but 3.4 Å for the refinement. A choice was made to truncate the data for refinement based on the anisotropy of the data and the low completeness in the high resolution shells.

Page 8 figure 3C: it is hard to see the relative orientation of CirpA1 on the surface of the truncated prepared in.

This comment is not clear to us. Fig 3C has the properdin orientated identically in the two sub-panels in order to illustrate the two different interaction surfaces used by the CirpA1 in the crystal.

Page 8 figures 3D: why is the y-axis using absorbance units rather than percent lysis as shown in the assays in figure 1 and figure 2? It would be really beneficial for this manuscript if the same assay parameters represented in all of the figures, particularly because these assays are comparing *E. coli* expressed protein compared to the previous studies which uses S2 expressed material. It would be very interesting to know how well they compare.

The intention of this figure is to compare the three CirpA1 variants, i.e. this is showing a relative effect. We agree on the utility of comparing S2 and *E. coli* produced material and have included this as Supplementary figure 5.

Page 7, last paragraph: in order to demonstrate further the importance of the interface between properdin and CirpA1, it would be really good to also test the residues that comprise the properdin binding interface using point mutations in similar to what was performed on CirpA1.

Again, we apologize, our capacity to carry out these experiments was impacted by the UK lockdown starting in March 2020, and by the subsequent move of the lab to the USA.

Page 9, line 25: there's a comment that there is high structural similarity between OMCL and CirpA1. This should be extended to the sequence identity the sequence identity.

We have added the degree of structural similarity (r.m.s.d. of 1.4 Å over 105 C α s) and toned down the language.

Page 10 figure 5 a: this is a little bit confusing to the non-complement expert reader because in figure 1 you go to show that the inhibitor only inhibits the alternative pathway but in this diagram you have CirpA1 depicted as inhibiting C3 convertase only at the amplification step. Diagram implies that there is no inhibition of the upstream initiation pathways.

As the reviewer correctly points out in earlier comments, the CirpA1 inhibition of properdin should impact all three pathways via the amplification loop. We attempted to illustrate this in this diagram. We interpret the pathway specific effects seen in figure 1 as being a function of the assay system used.

Page 10 line 26: the authors do allude to the tech species using cattle. As hosts they should be expanded upon in the introduction and should be incorporated into the lysis essays in figure 2 for a complete story.

As mentioned above, the focus of this study was to identify potential therapeutics for humans, and the potential importance of ungulates was introduced into the discussion at the time of writing the manuscript. We agree it would be interesting to probe species specificity of the CirpA family further in future work.

Page 11 second paragraph: Have the wt CirpA1, and mutants of CirpA1, been tested on the wild type properdin and imaged using the ultrastructure such as using Cryo EM to correlate binding sites on properdin? How feasible would this be. It would really make the research much structure.

The small, thin domains of properdin, combined with flexibility, make it a challenging target for Cryo-EM. To date it has only been possible to image it at sufficient resolution to see an extra 15kA protein when it is bound to C3bBb. As CirpA prevents this interaction this rules out this route.

Page 11 line 22: the production of C3b was not described in the methods. How was C3b made?

C3 was purchased from CompTech, USA, and activated to C3b using the protocol described in Berends et al 2015, BMC Biol. We have added more detail to the "Competition Assay" section where this was described.

Methods: the writing of the message looks rather rushed and needs to be fixed up with respect to scientific nomenclature such as the use of subscripts on chemicals italics for species, the source of important things such as antibodies and C3b.

We apologize and have corrected the methods section.

Extended data: there are no legends for the two figures that comprise the extended data.

Supplementary data has been expanded in response to the reviewers comments and now has legends.

First first extended data figure: part C: this table doesn't really make much sense What's the point of having a blast searches if it was already known that OmCI was a homologue. Is OmCI in this table?

The table summarizes the top BLAST hits for each of the seven proteins identified in the complement inhibitory fraction of the fractionation procedure. Only AP1 (CirpA1) was subsequently shown to be a complement inhibitor. OmCI is not in the table as it is only identified when searching against the PDB, and even then only with the FFAS algorithm, as the hit is very weak.

Reviewer #3 (Remarks to the Author):

Review of the manuscript 'Targeting properdin - Structure and function of a novel family of tick-derived complement inhibitors' by Dr. Susan lea and colleagues.

In this work, the authors identified and functionally characterized a new family of a complement-regulator from the saliva of the tick *Rhipicephalus pulchellus*.

Complement biology is experiencing a bout of renewed interest. Not only because it is becoming increasingly clear that complement dysfunction contributes to a larger range of human diseases than previously thought but also because complement is emerging as a major culprit in the pathogenesis of severe COVID-19.

Early complement-targeting strategies has not been as successful as initially hoped for. This is in part due to the facts that complement is overall more complex in its effects on other immune sensors/effector systems than previously thought, that it also exerts (immune) homeostatic and tissue repair functions, and is operative at unexpected locations well beyond the vascular space. Thus, new tools allowing to uniquely modify this system will be helpful. Properdin is the only known positive regulator of complement. Properdin stabilizes the C3 convertase and thus supports prolonged and more effective alternative pathway activation. There is thus clinical interest to target properdin and several blocking anti-properdin Abs have been developed.

Here, the group of Dr. Lea has identified the first 'natural' properdin inhibitor from a tick species. This protein, CirpA1, a member of the lipocalin family proteins, binds properdin and leads to potent inhibition of AP activity. The group further identify CirpA homologues from other tick species, present crystal structures of these (in part with essential properdin domains) and use these to elegantly demonstrate why the different CipA family members display AP control host serum species specificity.

The work is important and timely and all experiments are solidly performed with data interpretations accurate.

We thank the reviewer for the kind comments.

Experimentally, I could only argue that a set of Biacore experiments using purified components

of AP C3 convertase formation and CirpA1 would nicely complement the data presented in Figure 1f (and exclude any serum-derived unknown component that may have impact).

We did attempt to carry out SPR and ITC analysis of the CirpA-properdin interaction, but were unable to produce data that could be satisfactorily fit, due to a combination of self stickiness and tendency to interact with the SPR chip surface. We then attempted to use microscale thermophoresis (MST) and were able to produce data using fluorescently tagged properdin (data attached), though not with the label on the CirpA. The data was fit to give a K_d of ~100nM, which is 5-10 fold weaker than the apparent K_d inferred from the IC50 measured in the complement inhibition assays. We concluded that the labelling procedure perhaps led to some degree of steric inhibition of the binding, but we can include this data in the manuscript if the reviewer feels it will be helpful.

Few minor comments:

1. Although mentioned in the title, the abstract should also contain the information that CirpA1 was identified in the tick.

Corrected.

2. The labeling of the figures (particularly Figure 1 with ve- and ve+ denoting different conditions in different sub panels; OmCl is mentioned in the text but not legend, etc.) is really confusing and could be more intuitive. I would suggest to make generally label panels better (for example, Figure 2a could have the tick species connected to the CirpA proteins, etc. etc.).

We apologise and have added extra information to the Figure 1 legend to help the reader. We decided not to include the tick species in Fig 2a as the figure is already very busy, but we have included a table in Supplementary Figure 4.

3. Although data seem to be largely statistical significant ‘by eye’ (where statistical analyses are applicable), they should still be included (Fig 1c, 2b, 3d, etc.).

We have added statistical analysis to Fig 3d.

4. According to the complement nomenclature update 2019, properdin should be called ‘properdin’ and not FP.

We apologise and have corrected this throughout the manuscript.

5. The authors claim that their work represents the discovery of the first properdin inhibitor. It may be more accurate to mention that anti-properdin antibodies exist and work but that they identified the first ‘natural’ inhibitor.

We apologise for giving this impression, we tried to be careful with our language to stress that this is the first complete structural and functional analysis of a properdin inhibitor. In addition to the antibodies there is also a properdin inhibitor from *Ixodes scapularis* which we used as the

positive control in Figure 1f, but there is no structure available for this protein.

Reviewer #4 (Remarks to the Author):

This is an interesting and generally well-written manuscript describing the isolation from ticks of a protein inhibitor of properdin; binding of the tick protein prevents or reverses properdin binding to the amplification loop convertase and thus reduces convertase stability and complement activation downstream. Given the current huge interest in complement inhibiting drugs the work is of broad potential interest. The approach is not completely novel in that there are already several publications describing mAb against properdin that work in a similar manner and indeed these are progressing towards the clinic. Nevertheless, the thorough characterisation of a nature-derived inhibitor is important.

I have a few comments that I hope will improve clarity of the work:

1. A terminology issue (sorry!) - Factor P (FP) is not accepted terminology for properdin; the 2014 nomenclature report from ICS and ECN did not recommend change and even the 2019 opinion piece from Tenner and others (Front Immunol.2019 Jun 7;10:1308) that has gained traction in the field does not recommend changing the name. I am aware that others have used FP recently but given the likely influence of this article I really urge that the correct and approved nomenclature is used.

We appreciate the reviewer's perspective and have altered the terminology throughout the manuscript.

2. The statement in the introduction implies that the CP and LP directly lead to formation of the C3bBb convertase - although I know that this was not intended by the authors, the language is confusing - so just stress that this is the AP/amplification loop convertase.

Corrected.

3. Figure 1c; using the Weislab assay there is clear inhibition of the AP but NO inhibition of the CP/LP - given that the AP loop is the key amplifier for all pathways this is surprising and likely reflects an artefact of the assay. I would urge repeating using haemolysis assays for at least AP and CP.

CP haemolysis assays were attempted under a variety of conditions, with examples shown in the attached figure, but no inhibition of CP by CirpA1 was observed. This is consistent with previous observations of the difficulty of inhibiting the CP in haemolysis assays (eg, Harboe *et al*, Clin. Exp. Immunol. 2004).

4. Figure 1d/e measures C3a/C5a in the Weislab assay supernatants. The data would be more convincing if done outside of the assay with known AP activators - eg. zymosan - enabling dose response analyses.

We agree that further data would strengthen the observation, but would argue that the result of inhibition by CirpA1 in the assay is clear. The Wieslab AP assay does use a known AP activator on the plate (LPS).

5. Figure 2b shows the species specificity of the effect - it is surprising given its importance as a model that mouse was not included in this. Also, given later text on natural hosts, why not tested with bovine serum?

The focus of the study was identifying inhibitors with potential therapeutic relevance for humans. Hence the focus was initially on primates. We also tested other standard sera routinely kept in the lab for complement inhibition assays, though did not have access to mouse serum at the time these experiments were carried out. The potential importance of ungulates was introduced into the discussion at the time of writing the manuscript. We agree it would be interesting to probe species specificity of the CirpA family further in future work.

6. The crystal structures are very convincing as would be expected from these scientists; however, I don't understand why a truncated properdin is used rather than full-length. Does this not risk missing important interactions? Can the authors explain their reasons?

Many attempts were made to crystallize CirpA with full length properdin, both in the oligomeric and monomerized forms, but without success. This may be due to the potential for hinging between TSR domains observed in previous structures. Successful crystallization was only achieved with the delta-2,3 construct. While we can't rule out missing interactions with this

approach, the location of the CirpA1 binding site is at the opposite end of the molecule to the missing domains.

7. Species specificity is returned to in Discussion - are these ticks known to be primate parasites? It is stated that they have evolved to parasitise cows and other large domestic animals - this deserves attention as suggested above and perhaps by adding in these species to alignments.

We agree this warrants further study (see answer to point 5) and have altered the text to suggest this. However, we would prefer not to add to the alignment in Figure 4, as this represents species for which we have activity data.

8. The discussion around other lipocalins suggests other activities - equally possible that they have no relevant activities or that they do the same job in other species. I'm not sure that the comparison with the C5 inhibitor adds much (at least as currently discussed) and might confuse.

We apologise for the confusion and have re-written this section of the discussion. We believe that an analysis of the versatility of the lipocalin fold would be of interest to the general reader.

9. The absence of any direct measures of binding affinities using SPR or related methods is surprising - such data could aid understanding of how these proteins can not only prevent properdin binding to convertase but also displace from convertase.

We did attempt to carry out SPR and ITC analysis of the CirpA-properdin interaction, but were unable to produce data that could be satisfactorily fit, due to a combination of self stickiness and tendency to interact with the SPR chip surface. We then attempted to use microscale thermophoresis (MST) and were able to produce data using fluorescently tagged properdin (data attached), though not with the label on the CirpA. The data was fit to give a K_d of ~100nM, which is 5-10 fold weaker than the apparent K_d inferred from the IC50 measured in the complement inhibition assays. We concluded that the labelling procedure perhaps led to some degree of steric inhibition of the binding, but we can include this data in the manuscript if the reviewer feels it will be helpful.

10. Minor annoyances - subscript numbers in chemical names like Na_2PO_4 ; include degree symbol in temperatures like 25C etc.

We apologise and have corrected these throughout.

B.Paul Morgan

Reviewers' Comments:

Reviewer #1:

Remarks to the Author:

I am satisfied that the authors have reframed the interpretation of the results based on all of the reviewers comments. The complement pathway is complex! Overall this is an exciting result particularly in understanding of how properdin works as a stabilising factor in complement activation. Potentially there is also as a useful inhibitor or inhibitor design.

The methodology was sound and is now analysed more accurately. It is noted that there are potential experiments that should have been performed for addressing any folding issues but it is acknowledged that this could not be undertaken due to a lab move. Moving forward for future work then this should be automatically incorporated into the methodology where possible due to the conformation lability often observed in complement proteins.

Reviewer #3:

None

Reviewer #4:

Remarks to the Author:

The authors have done a reasonable job of addressing the reviewer comments - albeit with some unaddressed because of logistical issues related to Covid and relocation.

I have no further substantial concerns that would preclude publication of this interesting work.